# USP48 restrains resection by site-specific cleavage of the BRCA1 ubiquitin mark from H2A

Michael Uckelmann[1], Ruth M. Densham[2], Roy Baas[1], Herrie H.K. Winterwerp[1], Alexander Fish[1], Titia K. Sixma[1] & Joanna R. Morris[2]

BRCA1-BARD1-catalyzed ubiquitination of histone H2A is an important regulator of the DNA damage response, priming chromatin for repair by homologous recombination. However, no specific deubiquitinating enzymes (DUBs) are known to antagonize this function. Here we identify ubiquitin specific protease-48 (USP48) as a H2A DUB, specific for the C-terminal BRCA1 ubiquitination site. Detailed biochemical analysis shows that an auxiliary ubiquitin, an additional ubiquitin that itself does not get cleaved, modulates USP48 activity, which has possible implications for its regulation in vivo. In cells we reveal that USP48 antagonizes BRCA1 E3 ligase function and in BRCA1-proficient cells loss of USP48 results in positioning 53BP1 further from the break site and in extended resection lengths. USP48 repression confers a survival benefit to cells treated with camptothecin and its activity acts to restrain gene conversion and mutagenic single-strand annealing. We propose that USP48 promotes genome stability by antagonizing BRCA1 E3 ligase function.

[1] Division of Biochemistry and Cancer Genomics Centre, Netherlands Cancer Institute, Plesmanlaan 121, 1066 CX Amsterdam, The Netherlands. [2] Birmingham Centre for Genome Biology and Institute of Cancer and Genomic Sciences, Medical and Dental Schools, University of Birmingham, Birmingham B15 2TT, UK. Michael Uckelmann and Ruth M. Densham contributed equally to this work. Correspondence and requests for materials should be addressed to T.K.S. (email: t.sixma@nki.nl) or to J.R.M. (email: j.morris.3@bham.ac.uk)

To assure genomic integrity and protect against disease such as cancer, DNA double-strand breaks (DSB) need to be faithfully repaired. The cell can employ two major pathways to repair DSB, homologous recombination (HR), commonly thought of as error free, and the more error-prone non-homologous end joining (NHEJ). The choice between these two pathways is made at the point of DNA end resection[1]. Minimal processing directs repair to NHEJ whereas 5′-end resection in late S-phase and G2 directs repair to HR mechanisms including gene conversion (GC), which is the most accurate, and thus least mutagenic, form of DSB repair (reviewed in refs [2,3]). However, extensive resection can result in the use of a sub-pathway of HR repair known as single-strand annealing (SSA). In this process, extended resection reveals direct repeat sequences around the DNA breaks that can be annealed and the resulting single-stranded DNA (ssDNA) 'flaps' are processed to delete the material between the repeats (reviewed in refs [3–5]). SSA and GC compete for the repair of DSBs in budding yeast[6], but as SSA requires extended resection to expose direct repeats, limiting DNA end processing is critical to promoting accurate DSB repair. How the degree of end resection is controlled is not well understood.

Resection takes place over several defined phases. It begins by the endo- and then exonuclease activity of MRE11-CtIP, and is extended by Exo1 and BLM-DNA2 helicase/endonuclease complexes. The extensive ssDNA is bound by replication protein A (RPA), which is either exchanged for the recombinase RAD51, required for homology searching, strand invasion, and GC, or the RPA-bound sequences are annealed by RAD52 if homologous sequences are present in the resected ends[3–5].

BRCA1, the breast and ovarian cancer predisposition gene product, and 53BP1, the p53-binding protein, are opposing regulators to the degree of DNA end resection. In the absence of BRCA1, resection is blocked by 53BP1 and its effector proteins, promoting NHEJ and suppressing HR repair (reviewed in refs [7,8]). BRCA1 overcomes the 53BP1-mediated block through interaction with the resection protein CtIP[9] and through its E3 ubiquitin ligase activity[10]. The N termini of BRCA1 and BARD1 associate and establish an active E3 ubiquitin ligase[11].

The role of the ligase activity in BRCA1 function has been controversial. Although studies on the catalytically inactive Brca1-I26A mutant in murine cells suggest no effect on DNA repair[12,13], studies on RING-less Brca1 variants and on mice bearing a disruptive zinc-ligating variant of the Brca1 RING or entirely deleted Brca1 imply that ligase function may be relevant to genomic instability[14–17]. Moreover, the lethality of a Brca1 RING-less mouse strain is rescued by 53bp1 loss further supporting a possible interaction between ligase function and 53bp1[16].

Recently, a target of BRCA1 E3 ligase activity has been identified as a specific group of C-terminal lysines on H2A (K125/127/129)[18], thereby establishing a third specific ubiquitination site on H2A besides the previously identified sites K13/15 and K118/119, targets of RNF168 and polycomb repressor complexes 1 (PRC1), respectively[19–22]. Ubiquitination of H2A controls several aspects of the damage response: modification of K118/119 by PRC1 is thought to mediate DNA-damage-induced local transcriptional repression[23] as well to as potentiate the downstream signaling[24] resulting in 53BP1 accumulation, whereas ubiquitin conjugated at K13/15 directly acts to promote 53BP1 interaction at damaged chromatin[25]. Ubiquitination of H2A by BRCA1 promotes long-range resection and HR repair through the recruitment and activity of the Swr1-like remodeler, SMARCAD1, which repositions the 53BP1 block and permits resection[10].

Deubiquitinating enzymes (DUBs) are able to counteract ubiquitination by cleavage of the isopeptide bond between ubiquitin's C terminus and the target protein lysine. Several DUBs have been suggested to target H2A[26–30], but very little is known about their site specificity nor their role in the different repair pathways. We wanted to investigate whether DUBs that act on H2A-ubiquitin substrates show site specificity and whether known H2A DUBs would counteract the BRCA1-induced DNA damage response.

To find DUBs antagonizing BRCA1 E3 ligase activity, we tested a panel for site-specific H2A deubiquitination. In this analysis, USP48, previously identified as an interactor of ubiquitinated nucleosomes[31] but otherwise poorly characterized, appeared specific for the BRCA1 site and intriguingly needs an addition ubiquitin, which itself is not cleaved in *Cis* on the nucleosome, to be fully active. We show that in cells USP48 counteracts BRCA1 E3 ligase activity, restricting DNA end resection and RAD51 recruitment. Depletion of USP48 increases SSA and confers a RAD52-dependent survival benefit to cells treated with camptothecin. We propose USP48 as a regulator of the DNA damage response, counteracting BRCA1 E3 ligase activity. Moreover, we provide evidence that USP48 acts to prevent extensive resection and restrict the use of SSA.

## Results

**Assessing site specificity of H2A DUBs.** Each of the E3 ligases that modify H2A specifically monoubiquitinates distinct groups of lysines. K125/127/129 are ubiquitinated by BRCA1-BARD1 (H2A$^{BRCA1ub}$), K118/119 by PRC1 complexes (H2A$^{PRC1ub}$), and K13/15 by RNF168 (H2A$^{168ub}$). This site specificity is retained in vitro[18,21,22] and allows reconstitution of ubiquitinated nucleosomes. To address whether DUBs specific for these three sites exist, we selected a subset, previously suggested to deubiquitinate H2A (USP3, USP16, BAP1/ASXL1[26–30]) and/or involved in the DNA damage response (USP1/UAF1, USP11, USP7,USP15, USP12/UAF1)[32]. In addition, we included USP48, because it has been identified as a potent binder of ubiquitinated nucleosomes[31].

We produced the DUBs recombinantly and purified them for biochemical characterization (Supplementary Fig. 1a, b). To assess basal activity we examined the minimal substrate ubiquitin–rhodamine (Ub$^{Rho}$), where we followed increase of fluorescence intensity upon cleavage of a quenched Rhodamine-labeled peptide. Full kinetic analysis showed that all DUBs are active (Supplementary Fig. 1a and Supplementary Table 1).

We then tested for site-specific cleavage of ubiquitinated nucleosome core particles (NCPs) where a drop in the fluorescence polarization (FP) signal of TAMRA-labeled ubiquitin ($^{TAMRA}$Ub) indicates cleavage (Fig. 1a). In this assay, most DUBs have no clear preference for any of the three different nucleosome ubiquitination sites (Supplementary Fig. 1c and d). The exception was USP48, which has a preference for H2A$^{BRCA1ub}$ over H2A$^{168ub}$ and H2A$^{PRC1ub}$. USP48 was also the most active of the DUBs tested on a nucleosomal substrate, where we needed to lower the DUB concentration 10-fold (to 50 nM), to allow adequate readout in the FP assay. Furthermore, when we assessed whether a nucleosomal substrate is preferred over the minimal substrate, by normalizing the observed rates from the FP assay to the activity of the respective DUBs on minimal substrate (Supplementary Fig. 1e and Fig. 1b), we found USP3 and USP48 prefer the nucleosomal substrate.

**USP48 is specific for H2A$^{BRCA1ub}$.** Analysis of USP48 cleavage time courses examined by western blotting confirmed the observed preference for H2A$^{BRCA1ub}$ over H2A$^{168ub}$ or H2A$^{PRC1ub}$ (Fig. 1c). This analysis surprisingly revealed that

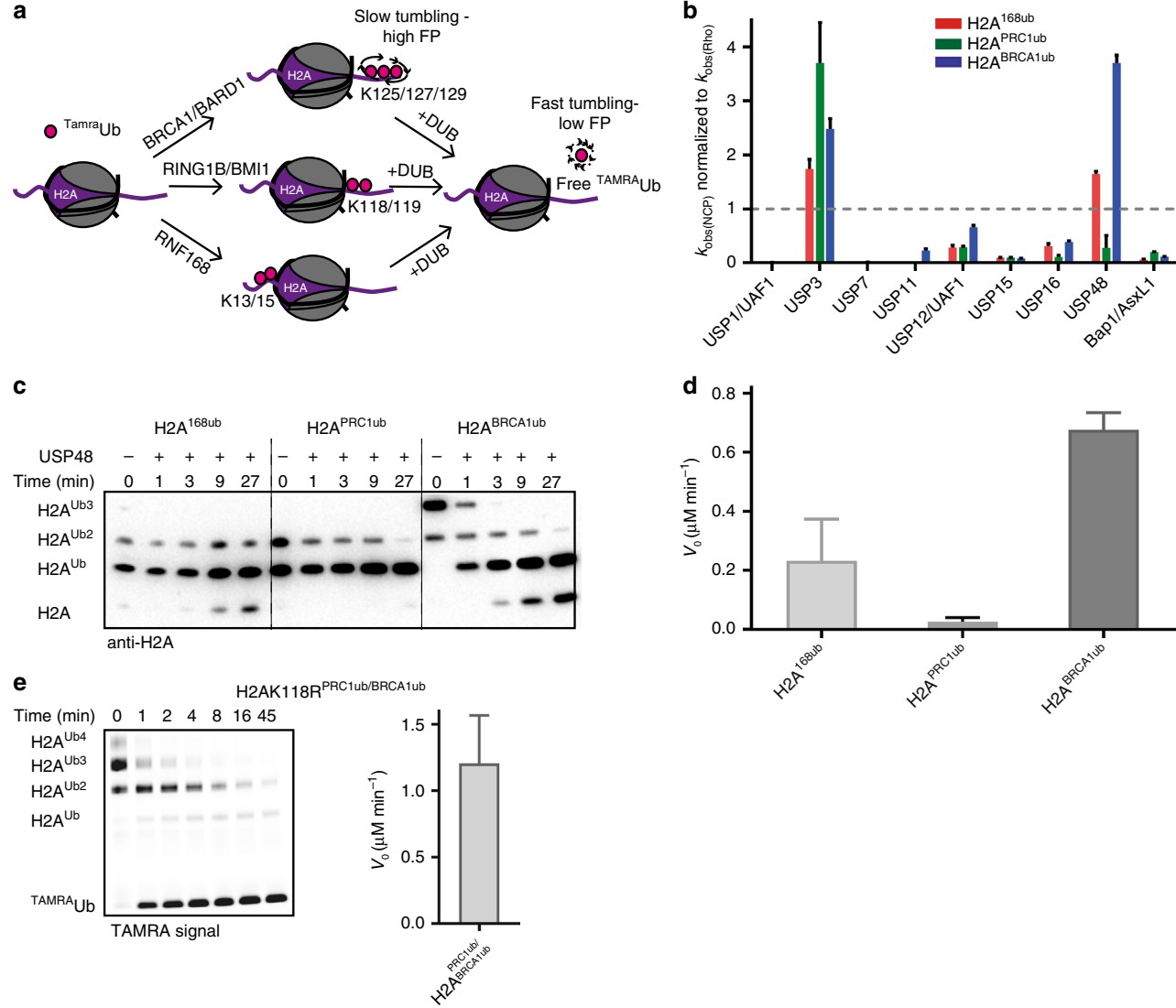

**Fig. 1** USP48 specifically deubiquitinates H2ABRCA1ub. **a** Schematics of the fluorescence polarization screen. Recombinant nucleosome core particles are site specifically ubiquitinated using the E3 ligase named and $^{TAMRA}$Ub. Upon addition of a DUB, cleavage can be followed by a decrease in FP signal. **b** USP48 prefers nucleosomal substrates. Site-specific cleavage of H2A$^{168ub}$, H2A$^{PRC1ub}$, and H2A$^{BRCA1ub}$ in relation to activity on minimal substrate. Observed $k$-values from fitting the raw data of the FP assay ($k_{obs(NCP)}$) (Supplementary Fig. 1c) were normalized to observed $k$-values on minimal substrate ($k_{obs(Rho)}$) obtained from fitting an exponential function to the traces in Supplementary Fig. 1e). A value above one indicates that NCPs are the preferred substrate. The mean of two technical replicates ± SEM is shown. **c** Time course of USP48 cleavage of H2A$^{168ub}$, H2A$^{PRC1ub}$, and H2A$^{BRCA1ub}$, anti-H2A western blotting. USP48 only cleaves efficiently when more than one ubiquitin is present on the BRCA1 site. The blot shown is representative of two experiments. **d** USP48 cleaves multi-monoubiquitinated H2A$^{BRCA1ub}$ faster than multi-monoubiquitinated H2A$^{168ub}$ and H2A$^{PRC1ub}$. Quantification of the initial reaction velocity ($V_0$) of USP48 measured by the liberation of free $^{TAMRA}$Ub. Gels used for quantification are shown in Supplementary Fig. 2a. The mean of two technical replicates ±SEM is shown. **e** USP48 does not cleave H2A$^{PRC1ub}$ when H2A$^{BRCA1ub}$ is present, but all H2A$^{BRCA1ub}$ is cleaved. Left panel: cleavage of NCPs, ubiquitinated on PRC1 site with unlabeled ubiquitin, and on the BRCA1 site with $^{TAMRA}$ub by USP48 was followed on gel. The TAMRA fluorescence is used as readout. Right panel: quantification of the initial reaction velocity ($V_0$) measured by liberation of free ubiquitin. The mean of two technical replicates ± SEM is shown

USP48 cleaves H2A$^{BRCA1ub}$ most efficiently if H2A is ubiquitinated on more than one lysine (H2A$^{BRCA1ub3}$, H2A$^{BRCA1ub2}$). To quantify the activity we analyzed cleavage time courses of H2A$^{BRCA1ub}$, H2A$^{168ub}$, or H2A$^{PRC1ub}$ on gel using fluorescence of tetramethylrhodamine $^{(TAMRA)}$Ub as readout (Supplementary Fig. 2a). Comparison of the initial linear reaction rates revealed that H2A$^{BRCA1ub}$ is cleaved with an activated rate, compared with H2A$^{168ub}$ and H2A$^{PRC1ub}$ (Fig. 1d). We conclude that USP48 shows a preference for H2A$^{BRCA1ub}$ and cleaves its substrate efficiently if more than one ubiquitin is present on the

site. We will refer to the additional ubiquitin needed for activity as the 'auxiliary' ubiquitin, because it aides USP48 cleavage but itself does not get cleaved.

We wondered how the observed preference for H2A$^{BRCA1ub}$ translates to a situation where multiple ubiquitination sites are available for cleavage on a single H2A. To test this we generated NCPs with an unlabeled ubiquitin conjugated to the polycomb site and a $^{TAMRA}$Ub conjugated to the BRCA1 site. As expected, we find that USP48 cleaves these substrates with the activated rate (Fig. 1e). The presence of a ubiquitin on H2A$^{PRC1}$ seems to

further accelerate the cleavage of H2A$^{BRCA1ub}$. Interestingly, no H2A monoubiquitinated with $^{TAMRA}$Ub is observed, showing that unlabeled ubiquitin on the PRC1 site is not cleaved when a labeled ubiquitin on the BRCA1 site is available. Moreover, on this substrate all the H2A$^{BRCA1ub}$ is cleaved, suggesting USP48 is capable of cleaving all H2A$^{BRCA1ub}$ modifications when an auxiliary ubiquitin is placed elsewhere on H2A.

We next analyzed potential cleavage of di-ubiquitin linkages, using USP16 as a positive control (Supplementary Fig. 2b). Of all the linkages only K27-linked di-ubiquitin was cleaved by USP48. Moreover, the amount of cleaved di-ubiquitin after 45 min was negligible compared with the processing of ubiquitin from

modified nucleosomes. Taken together, our data suggest USP48 is specific for multi-monoubiquitinated H2A$^{BRCA1ub}$.

**An auxiliary ubiquitin on the nucleosome promotes USP48 activity**. To investigate the effect of the auxiliary ubiquitin on USP48 rates we performed substrate-binding assays combined with a detailed kinetic analysis and kinetic modeling using the software *KinTek Explorer*[33,34]. We quantified USP48-catalyzed cleavage of H2A$^{BRCA1ub}$ under different conditions, achieved by titrating either USP48 concentration, while keeping substrate concentration fixed or vice versa. Fig 2a, b show four conditions as an example in which the substrate concentration is kept steady

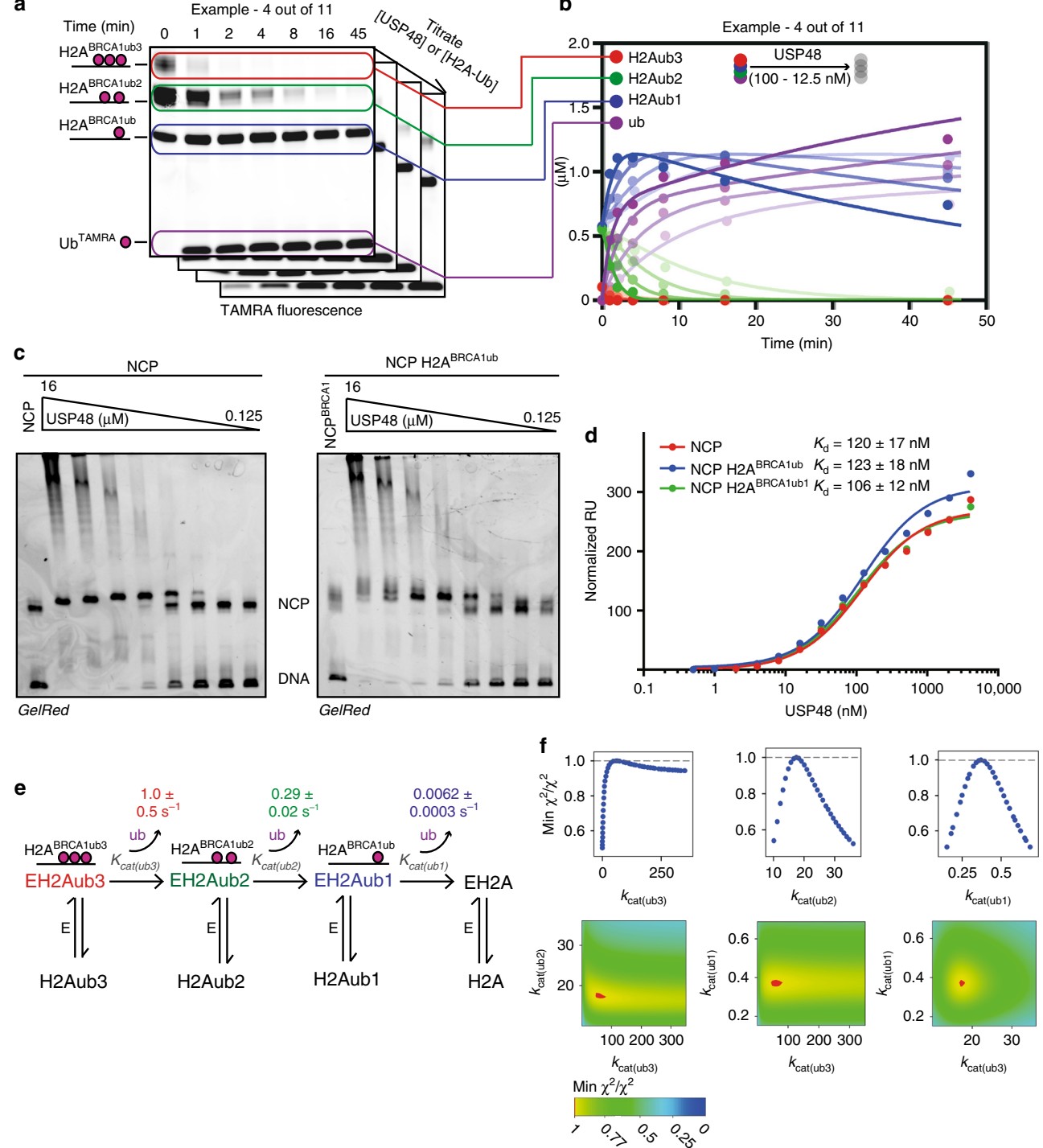

at 2 μM and USP48 concentration is varied between 100 and 12.5 nM. Supplementary Fig. 3 shows the whole range of conditions. We then tested affinities of USP48 for nucleosomes of different ubiquitination status and find that affinities are essentially the same for unmodified nucleosomes (NCP), monoubiquitinated nucleosomes (H2A$^{BRCA1ub1}$, generated by ubiquitinating K127 in a H2A$^{K125/129R}$ mutant) and multi-monoubiquitinated nucleosomes (H2A$^{BRCA1ub}$) in both gel-shift assays and surface plasmon resonance (SPR) experiments (Fig. 2c, d and Supplementary Fig. 3b). The gel-shifts show a distinct shift, and formation of higher molecular weight species at USP48 concentrations above 4 μM. We use the first distinct shift to estimate a Kd of 1 μM only, as the concentrations above 4 μM are never reached in our kinetic analysis and the effects may be due to artificial crowding. The SPR data indicate somewhat higher affinity, as we fit a $K_d$ of roughly 120 nM and therefore restrain binding between these values, in the fitting procedure that follows. Importantly, both SPR and gel-shift experiments show no difference in affinities for nucleosome species of different ubiquitination status, providing the rationale to link all binding constants for the kinetic modeling.

We fitted these binding and kinetic data to the simplest model that could describe the observed USP48 H2A deubiquitination pattern (Fig. 2e). We initially set the boundaries for the binding constants so they would reflect the upper and lower limits measured by gel-shift and SPR assays (1 and 100 nM, respectively). This initial fit determined 1 μM as the apparent $K_d$ for our model. We then fixed the binding constants of the model to 1 μM to fit the catalytic rates. The obtained best fit values for $k_{cat(ub3)}$, $k_{cat(ub2)}$, and $k_{cat(ub1)}$ describe the experimental data well (Fig. 2b for example and Supplementary Fig. 3 for all experiments) and are well constrained (Fig. 2f). The kinetic modeling indicates increased processivity when an auxiliary ubiquitin is present. The rates for $k_{cat(ub3)}$ (1 s$^{-1}$) and $k_{cat(ub2)}$ (0.29 s$^{-1}$) are similar, whereas $k_{cat(ub1)}$ is roughly 50 times slower (0.0062 s$^{-1}$). These results can be explained by a catalytic activation in the presence of the auxiliary ubiquitin or the inability of USP48 to cleave one of the three C-terminally ubiquitinated lysines efficiently.

To assess a possible role of the UBL domain of USP48 in the reaction we expressed and purified USP48 isoform 2 (uniprot identifier Q86UV5-2/USP48Iso2). The purity of the sample was similar for both isoforms (Supplementary Fig. 1b). USP48Iso2 lacked residues 909–962, which includes part of the UBL domain (Supplementary Fig. 4a). Full-length USP48 had significantly higher catalytic rates on both minimal substrate and H2ABR-CA1ub (Supplementary Fig. 4b, c and Supplementary Table 1). To compare the kinetic parameters of both isoforms on a nucleosomal substrate, we repeated the kinetic analysis outlined before with USP48Iso2 (compare Supplementary Fig. 4d, e and

Fig. 2e, f). The necessity for the auxiliary ubiquitin appears conserved in both isoforms, whereas catalytic rates are much higher for full-length USP48.

We further asked whether free ubiquitin could act as the auxiliary ubiquitin and increase USP48 processivity. To address this we performed Ub$^{Rho}$ and H2A$^{BRCA1ub}$ cleavage assays in the presence and absence of ubiquitin (Supplementary Fig. 3c, d). On both substrates the addition of free ubiquitin did activate USP48. This indicates that the auxiliary ubiquitin needs to be on the nucleosome and possibly in a defined orientation toward USP48, to cause activation (note that experiments in Supplementary Fig. 3d are done with USP48$^{Iso2}$ but both isoforms are expected to behave the same). In our analysis we regarded affinities measured by SPR and gel shifts (Fig. 2c, d) as actual substrate affinities. However, in order to engage the substrate (ubiquitin on H2A) a reorientation or conformational change of USP48 may occur on the nucleosome. This step might be masked in our assays as binding is dominated by the affinities for the nucleosome. Possible conformational change would be reflected in the calculated $k_{cat}$ values, which include the rate for catalysis and possible conformational change. Therefore, the detailed analysis does not yet fully assign the role of the auxiliary ubiquitin, but it firmly establishes that some form of activation takes place.

In summary, our biochemical analysis identified USP48 as a DUB specific for nucleosomes ubiquitinated by BRCA1-BARD1, identifying what is to our knowledge the first H2A DUB with in vitro specificity for the BRCA1 site. We also demonstrated the need of an auxiliary, nucleosomal, ubiquitin for USP48 to be fully active. We next sought to analyze USP48's importance in cells.

**USP48 deubiquitinating activity restrains resection.** First, we addressed whether USP48 could have a role in the DSB response by evaluating endogenous USP48 localization after DNA damage caused by micro-irradiation (IR). USP48 co-localized with 53BP1 and BRCA1 along the line of the laser, indicating recruitment to sites of DNA damage (Fig. 3a). Given the ability of USP48 to cleave H2A$^{BRCA1ub}$, we examined whether accumulation of USP48 at damage sites is dependent on BRCA1 and found that USP48 recruitment to damaged chromatin was markedly reduced in BRCA1-depleted cells (Fig. 3a, b).

The BRCA1-BARD1 E3 ubiquitin ligase activity promotes chromatin remodeling at damaged sites through H2A$^{BRCA1ub}$ modification, which consequently promotes long-range resection[10], RAD51 foci formation, and GC[10]. Using EdU incorporation and staining to identify S-phase cells, we examined the role of USP48 in resection and GC in irradiated cells. USP48-depleted U2OS and HeLa cells exhibited greater numbers of RAD51 foci formation after IR compared with controls (Supplementary

**Fig. 2** USP48 cleavage rates on H2ABRCA1ub are modulated by an auxiliary ubiquitin. **a** An auxiliary ubiquitin activates USP48 processivity. Gel-based assay to measure USP48 cleavage of H2A$^{BRCA1ub}$. $^{TAMRA}$Ub was used as readout. Cleavage was recorded under several different substrate and enzyme concentrations. Four (stacked) gels are shown as an example (Supplementary Fig. 3 for all gels). **b** Quantification of (**a**. Four out of 11 different conditions shown as an example (Supplementary Fig. 3 for full panel). Solid lines show fit obtained by global fitting of all 11 quantified time courses and binding data to the model defined in **e** using *Kintek Explorer*. **c** USP48 binds with similar affinities to ubiquitinated and unmodified NCP. USP48 is titrated to 50 nM of unmodified NCP or NCP$^{BRCA1ub}$ and analyzed by native gel-shift assays. A Kd of 1 μM is estimated. **d** USP48 binds nucleosomes of different ubiquitination status with the same affinities. Binding of USP48 to H2A$^{BRCA1ub}$, H2A$^{BRCA1ub1}$, and unmodified nucleosomes measured by surface plasmon resonance. Nucleosomes were immobilized on the surface. Normalized RU values of 10 successive injections of different USP48 concentrations are fitted by a one-phase binding model using GraphPad Prism (raw data in Supplementary Fig. 3b). $K_d$ and SE of the fit are indicated. **e** USP48 cleaves H2A$^{BRCA1ub}$ in nucleosomes 50–150 times faster than when auxiliary ubiquitin is present. Kinetic model describing USP48's cleavage pattern on H2A$^{BRCA1ub}$ with the fitted values for $k_{cat(ub3)}$, $k_{cat(ub2)}$, and $k_{cat(ub1)}$, and the SE of the fit. **f** Parameters for $k_{cat(ub3)}$, $k_{cat(ub2)}$, and $k_{cat(ub1)}$ in **e** are well constrained by the data. Evaluation of the goodness of fit. The upper three panels show how well defined the lower and upper boundaries are for the individual parameters. $k_{cat(ub1)}$ and $k_{cat(ub2)}$ fall into a well-defined local $\chi^2$ minimum. For $k_{cat(ub3)}$, the lower boundaries are well defined which allows the conclusion that $K_{cat(ub3)}$ should always be faster than $k_{cat(ub1)}$. The lower panel shows how $\chi^2$ varies when two of the fitted variables are varied against each other. Red indicates a $\chi^2$ minimum

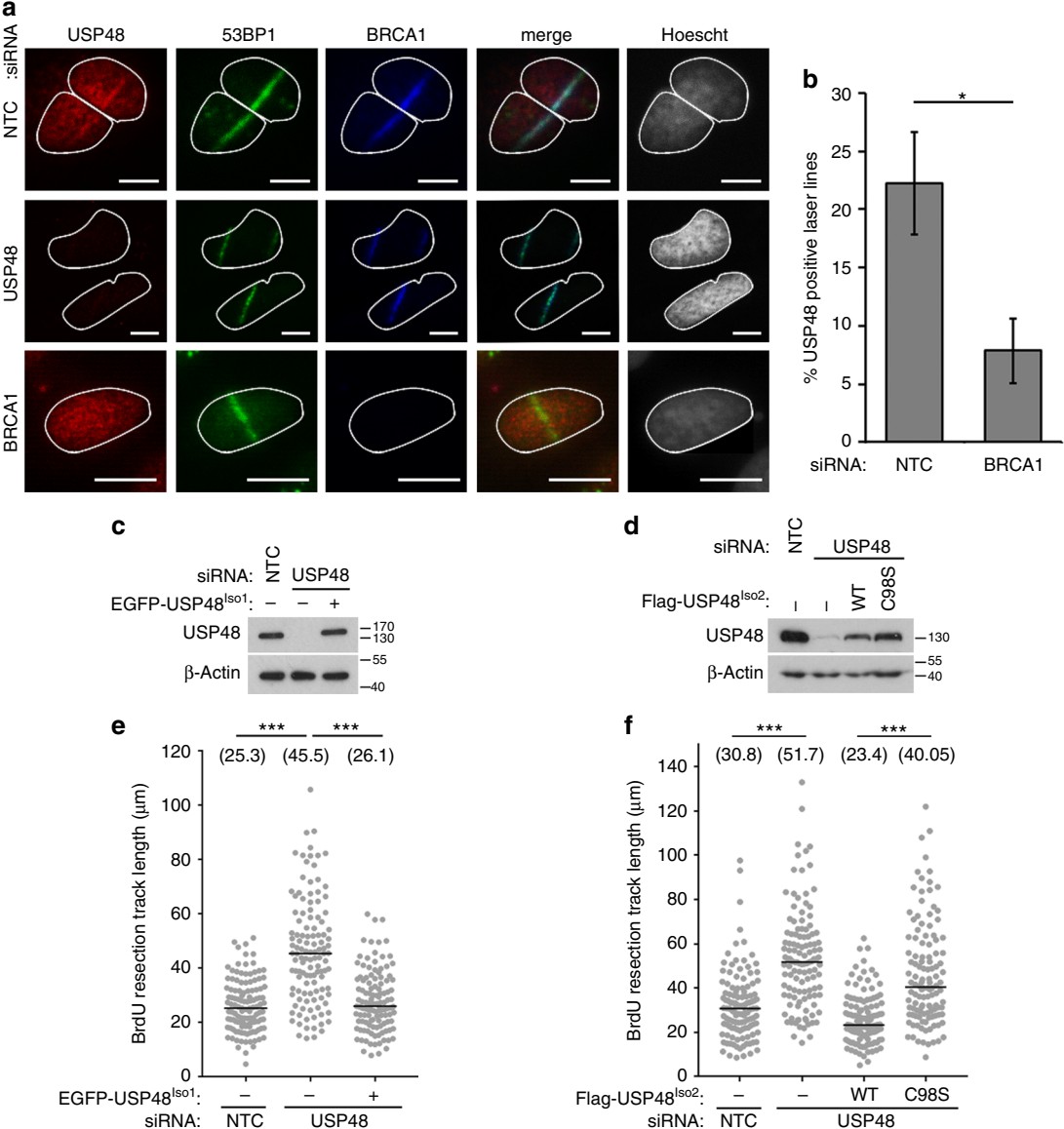

**Fig. 3** USP48 deubiquitinase activity restrains resection. **a** USP48 is recruited to sites of damage. BrdU-sensitized HeLa cells were subjected to laser stripe IR followed by fixation after 30 min and staining for endogenous USP48, BRCA1, and 53BP1. Scale bar = 10 μm. In the middle and lower panel, cells were treated with USP48 (exon 5 and exon 11 targeting sequences) and BRCA1 siRNA, respectively. USP48 siRNA treatment utilized exon 5 and exon 11 targeting sequences together throughout the manuscript, unless otherwise stated. **b** Percentage of USP48-positive laser lines in control vs BRCA1-depleted cells. Graph is mean from three independent experiments. Error bars are SEM and *$p < 0.05$ Student's *T*-test. **c, d** Expression levels of USP48 in stable HeLa-FlpIn cell lines following depletion of USP48 and DOX-induced expression of siResistant EGFP-USP48$^{Iso1}$ **c** or Flag-USP48$^{Iso2}$-WT or Flag-USP48$^{Iso2}$-C98S **d**. **e, f** USP48 depletion increases resection lengths. Resection lengths were measured in HeLa cells depleted for USP48 and complemented as in **c** and **d**. Cells were incubated with 10 mM BrdU for 24 h with addition of 10 μM Olaparib for the final 16 h. Cells were lysed and DNA fibres spread before staining for BrdU-positive single-stranded DNA resection tracks. One hundred and twenty fibres were measured using ImageJ for each condition. Bars indicate median, also shown numerically in brackets. ***$p < 0.005$ Mann–Whitney test

Fig. 5a, b). Moreover, RAD51 foci were brighter in USP48-depleted cells (Supplementary Fig. 5c), suggesting more RAD51 molecules per foci as well as greater detection of numbers of foci. We next assessed the relative contribution of the different USP48 isoforms. Complementation with either small interfering RNA (siRNA)-resistant wild-type (WT) USP48$^{Iso1}$ or USP48$^{Iso2}$ restored RAD51 foci numbers to control levels, suggesting that despite the difference in catalytic rates both isoforms are capable of regulating RAD51 foci formation (Supplementary Fig. 5d, e). In contrast, complementation with the catalytic mutant form of USP48$^{Iso2}$ where the catalytic cysteine was mutated to serine

(USP48$^{Iso2}$-C98S) was unable to restore RAD51 foci numbers to control levels (Supplementary Fig. 5e).

The ssDNA-binding protein RPA can be used as a marker of resection before RAD51 exchange, strand invasion, and GC. Irradiated S-phase cells depleted of USP48 had more RPA foci compared with controls (Supplementary Fig. 5f, g) and complementation with either siRNA-resistant USP48$^{Iso1}$-WT or USP48$^{Iso2}$-WT, but not USP48$^{Iso2}$-C98S, restored RPA foci numbers to control levels (Fig. 3c, d and Supplementary Fig. 5f, g). To assess resection itself, we incubated cells with Bromodeoxyuridine (BrdU) and treated with olaparib before measuring the

DNA track lengths of the exposed BrdU epitope, indicating resected ssDNA[10,35]. Median lengths of ssDNA were longer in USP48-depleted cells compared with controls (Fig. 3e, f). Moreover expression of siRNA-resistant USP48[Iso1]-WT or USP48[Iso2]-WT, but not USP48[Iso2]-C98S, restored resection to control lengths (Fig. 3e, f). Together, these data suggest that the catalytic function of USP48 acts to restrain DNA resection lengths at sites of damage.

**USP48 antagonizes the BRCA1-H2Aub-SMARCAD1 resection pathway**. Remodeling of chromatin-associated 53BP1 adjacent to

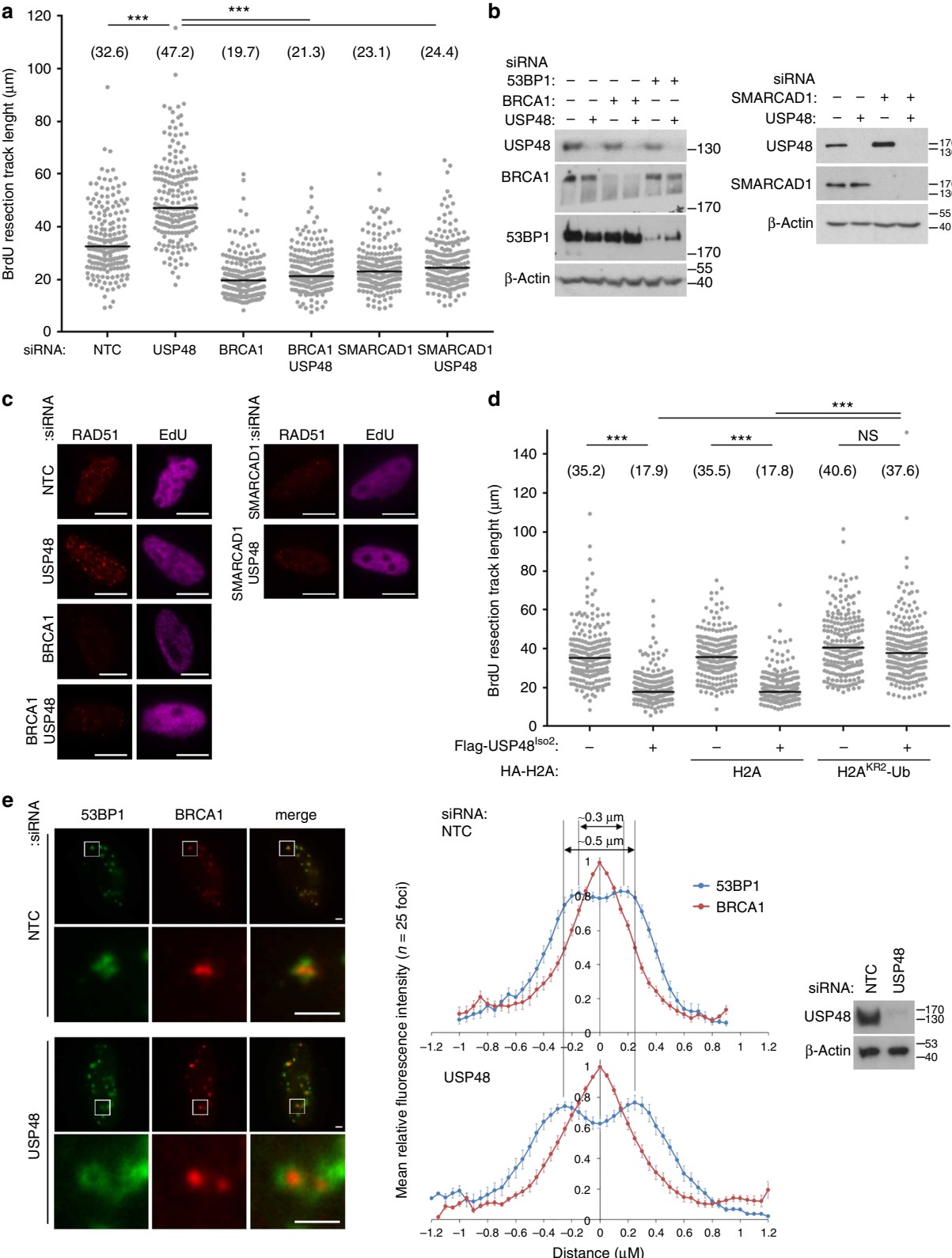

DNA DSBs by the BRCA1-BARD1 E3 ubiquitin ligase is mediated by the SWI/SNF-related helicase SMARCAD1[2]. To address whether USP48 acts in this pathway we tested whether the increased resection lengths observed on USP48 loss required BRCA1 or SMARCAD1. As expected[10], loss of BRCA1 or SMARCAD1 shortened ssDNA lengths (Fig. 4a, b) and, significantly, these shortened resection tracks were unaffected by USP48 co-depletion (Fig. 4a). These findings are consistent with the idea that USP48 loss has little impact if there is an absence of the ubiquitin mark to be cleaved or a shortage of the 'reader' to interpret that ubiquitination mark.

Consistent with a model in which the consequences of USP48 loss requires both BRCA1 and SMARCAD1, RAD51 foci numbers were severely decreased in cells co-depleted for BRCA1, or SMARCAD1 with USP48 (Fig. 4c, quantified in Supplementary Fig. 6a, b). To test BRCA1-dependency further we examined Rad51 foci in mouse embryonic fibroblasts (MEFs) bearing a homozygous deletion of murine Brca1 exon 11 ($Brca1^{\Delta 11/\Delta 11}$). In WT MEFs, Rad51 foci numbers were increased following murine Usp48 depletion compared to controls, while in $Brca1^{\Delta 11/\Delta 11}$ MEFs, which exhibit reduced Rad51 foci numbers[36], Usp48 depletion had no effect on Rad51 foci formation (Supplementary Fig. 6c, d). These data are consistent with a requirement for Brca1 driving increased resection and subsequent Rad51 loading in the absence of Usp48.

Previous analysis has revealed that over-expression of DUBs capable of catalyzing the removal of K118/K119 or K13/15-ubiquitin from H2A, results in disrupted 53BP1 and BRCA1 accumulation[30,37]. Although we observed that USP48 has little activity on alternate H2A ubiquitination sites in vitro, we could not exclude the possibility that USP48 processes these adducts in cells. However, ectopic expression of USP48[Iso1] had no effect on the ability of cells to form 53BP1 or BRCA1 foci following IR (Supplementary Fig. 7a, b). These data suggest that USP48 is not readily able to disrupt damage-signaling dependent on the H2A[168ub] modification nor, as loss of the PCR1 mark disrupts BRCA1 and 53BP1 recruitment[24], on the H2A[PRC1ub] modification.

We[10] and others[17] have noted that expression of H2A with ubiquitin genetically fused to the C-terminus can restore measures of GC in BRCA1 or BARD1-deficient cells. Such a fusion lacks an isopeptide bond and is resistant to DUBs. To test the hypothesis that USP48 acts to cleave ubiquitin from the C-terminus of H2A in cells we measured the impact of USP48 over-expression on resection lengths in cells expressing control H2A and H2A-Ub fusions, labeled H2A[KR1]-Ub and H2A[KR2]-Ub (see Supplementary Fig. 7c, d legend for details). Note previous analysis has shown ectopic H2A and H2A[KR2]-Ub are incorporated into chromatin[10]. Ectopic expression of USP48[Iso2] resulted in shorter ssDNA lengths (Fig. 4d), consistent with the enzyme's ability to restrict resection. However, USP48-induced shortening of resection lengths was prevented by expression of the H2A[KR2]-

Ub fusion, but not by H2A (Fig. 4d). Similarly, normal levels of RAD51 foci were restored by co-expression of either H2A[KR1]-Ub or H2A[KR2]-Ub fusions, but not H2A, in cells over-expressing USP48[Iso1] (Supplementary Fig. 7c, d). These data indicate that a protease-resistant H2A-Ub renders resection and subsequent RAD51 foci formation insensitive to the impact of USP48 overexpression, consistent with the enzyme's function in cleaving C-terminal H2A modifications.

BRCA1-BARD1 ligase function contributes to the positioning of 53BP1 away from the core of IR-induced foci (IRIF)[10]. To test the impact of USP48 on 53BP1 positioning, we measured the distribution of 53BP1 in BRCA1-associated foci in USP48-depleted cells and observed a greater spread of 53BP1 accumulations compared to controls (half peak intensity width ~ 1.1 μm in USP48-depleted cells compared with ~ 0.8 μm in controls) and a larger 'hole' at the foci core (53BP1 peak-to-peak distance of ~ 0.5 μm USP48-depleted cells, compared to ~ 0.3 μm in controls Fig. 4e). Thus, consistent with its relationship with BRCA1, SMARCAD1, H2A-Ub, and resection, USP48 appears to constrain the extent of 53BP1 repositioning at IRIF.

**USP48 restrains HR repair mechanisms.** DNA end resection is the decision point that commits cells to DSB repair by HR-mechanisms and not NHEJ. Incomplete resection results in reduced NHEJ and IR-sensitivity, which can be rescued by inhibiting resection initiation by repressing CtIP[38]. To assess the role of USP48 in NHEJ vs. HR, we measured IR sensitivity of USP48-depleted cells and their ability to reconstitute green fluorescent protein (GFP) in a NHEJ cut substrate (Supplementary Fig. 8a). USP48-depleted cells showed a slight reduction in NHEJ and slightly increased sensitivity to IR (Supplementary Fig. 8a–c). These small effects were lost when CtIP was co-depleted (Supplementary Fig. 8d, e), suggesting that the IR-sensitivity seen in USP48-depleted cells is due to resection. This minor resection-dependent NHEJ impairment suggests USP48 contributes only in a small way to the decision between HR-mediated repair and NHEJ.

USP48 loss increases resection and intriguingly, depletion of USP48 or 53BP1 or their co-depletion, all result in a similar increase in RAD51 foci (Supplementary Fig. 8f), suggesting that loss of either protein has a similar impact and acts in the same pathway to restrict RAD51 accumulation. We therefore assessed the role of USP48 in GC and SSA-mediated repair. Using the GC DR3-reporter assay, we measured the formation of GFP products from the integrated substrate in USP48-depleted cells transfected with I-Sce1[39], and complemented with USP48[Iso2]-WT or USP48[Iso2]-C98S (Fig. 5a). GC was increased in USP48-depleted cells and in USP48[Iso2]-C98S-complemented cells. These data show that USP48 activity restrains GC repair. To measured SSA-mediated repair we used an integrated reporter construct that depends on extensive resection and SSA for generation of a

**Fig. 4** USP48 antagonizes BRCA1-mediated resection. **a** Increased resection seen on USP48 knockdown requires BRCA1 and SMARCAD1. Resection lengths were measured in HeLa cells depleted as indicated. Cells were treated with 10 mM BrdU for 24 h with addition of 10 μM Olaparib for the final 16 h. Cells were lysed and DNA fibres spread before staining for BrdU-positive single-stranded DNA resection tracks. $n = 190$ tracks for each condition. Bars indicate median, also shown numerically in brackets. ***$p < 0.005$ Mann–Whitney test. **b** Western blottings to demonstrate protein expression levels in HeLa cells following siRNA as indicated. **c** Rad51 foci formation in S-phase (EdU positive) HeLa cells depleted as indicated. Cells were fixed at 2 h post 5 Gy IR. Scale bars = 10 μm (see Supplementary Fig. 6a, b for quantification). **d** Suppressive impact of USP48[Iso2] overexpression on resection lengths can be rescued by co-expression of an uncleavable H2A-Ub fusion. Resection lengths were measured in untransfected HeLa cells or those transfected with USP48[Iso2] and expressing either H2A or H2A-Ub fusion (KR2 denotes that lysines 13/15, 118/119, and 125/127/129 were mutated to arginines). Cells were prepared as in **a** and 210 tracks were measured for each condition. Bars indicate median, also shown numerically in brackets. ***$p < 0.005$, NS = nonsignificant, Mann–Whitney test. **e** USP48 restricts positioning of 53BP1 in damage foci. Images of BRCA1 and 53BP1 in cells treated with control or USP48 siRNA exposed to 2 Gy IR and fixed 8 h later. Scale bars = 2 μm. Quantification of mean relative intensity profiles for co-localizing foci. $n = 25$, bars = SEM. Right panel shows western blotting of USP48 protein levels

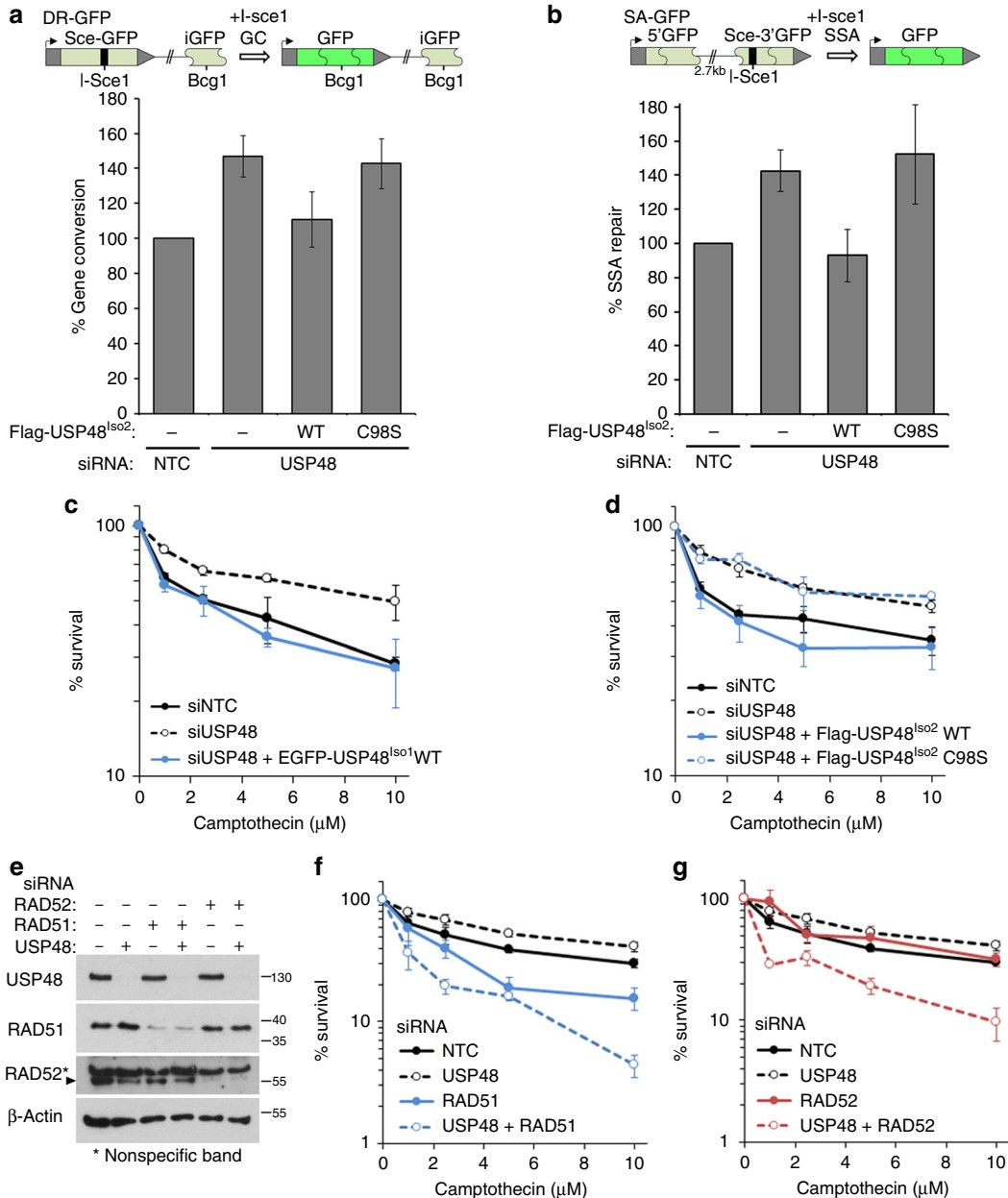

**Fig. 5** USP48 restrains homology-directed repair mechanisms. **a** Gene conversion (GC) was measured using U2OS DR3-GFP reporter cells as illustrated. It is noteworthy that the functional GFP cannot be produced by SSA from this substrate as the template iGFP lacks the 5'- and 3'-regions of GFP[38]. Cells depleted for USP48 were transfected with RFP, I-Sce1, and either Flag-USP48[Iso2]-WT or Flag-USP48[Iso2]-C98S. GFP-positive cells were normalized to RFP-transfection efficiency. %-repair is given compared with NTC. Graph shows mean, $n = 5$, error bars are SEM. **b** Single-strand annealing (SSA) was measured using U2OS SA-GFP reporter cells as illustrated. The two GFP fragments of the substrate share 266 nucleotides of homology. In principle GC with crossing over could also produce functional GFP; however, these have been shown to be rare events[39]. Cells were treated and analysed as in **a**. Graph shows mean, $n = 3$, error bars are SEM. **c**, **d** Camptothecin colony survival curves of HeLa cells depleted for USP48 and complemented with EGFP-USP48[Iso1]-WT, $n = 3$ **c** or Flag-USP48[Iso2]-WT or Flag-USP48[Iso2]-C98S, $n = 4$ **d**. Graph shows mean % survival normalized to untreated controls, error bars are SEM. **e** Western blottings to demonstrate USP48, RAD51, and RAD52 protein expression levels in HeLa cells following siRNA as indicated. **f**, **g** Camptothecin colony survival curves of HeLa cells depleted for USP48, RAD51, and both USP48 and RAD51 **f**, and USP48 and RAD52 alone or together **g**. Graph shows mean % survival normalized to untreated controls, $n = 3$, error bars are SEM

functional GFP (illustrated in Fig. 5b)[39]. USP48 depletion resulted in increased GFP expression (Fig. 5b), which was suppressed by complementation with USP48[Iso2]-WT, but not USP48[Iso2]-C98S. Moreover, both the increased GC and SSA repair seen in USP48-depleted cells required BRCA1, as well as CtIP (Supplementary Fig. 9a–c). Together, these data suggest that the role of USP48 DUB activity in restricting extended BRCA1-

mediated resection influences the outcome of HR at the level of both GC and SSA.

We next addressed the consequences of USP48 loss on cell survival following camptothecin exposure, as DSB lesions produced in this context rely on resection and HR for repair[40]. Remarkably, we found USP48-depleted or USP48[Iso2]-C98S-complemented cells were more resistant to camptothecin than

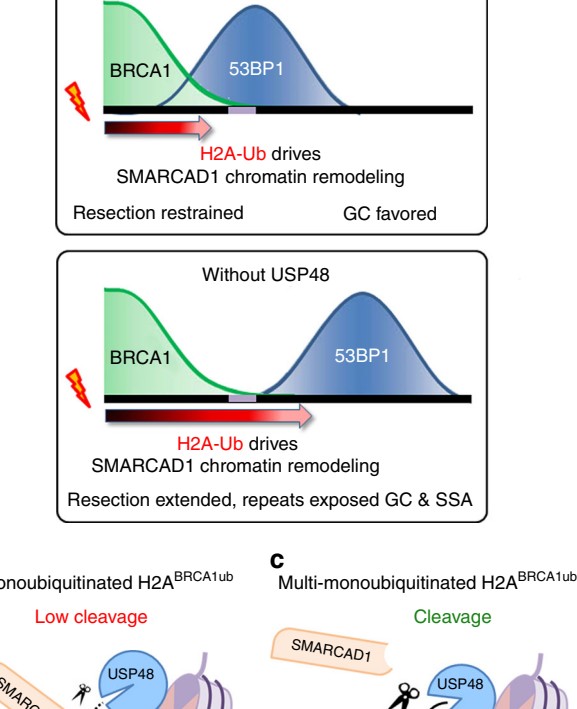

**Fig. 6** Model of USP48 function in restricting extended DNA end resection. **a** USP48 cleaves the C-terminal ubiquitin modification of H2A, limiting the extent of SMARCAD1 nucleosome remodeling and 53BP1 positioning. 53BP1 in turn restrains DNA end processing and consequently direct repeats are rarely exposed either side of the DSB, favoring GC (one side of the break is illustrated). Without USP48 activity, H2A modification is unopposed and SMARCAD1 nucleosome remodeling is extended further from the break, resulting in 53BP1 positioning further from the break site. 53BP1 is unable to constrain extended resection, resulting in increased GC, and, as repeats either side of the break are more often exposed, SSA is also increased. **b** USP48 cannot cleave BRCA1-initiated H2A monoubiquitination efficiently, therefore SMARCAD1 can bind and initiate downstream signaling. **c** A second ubiquitination on H2A can activate USP48, cleaving BRCA-initiated ubiquitination; therefore, SMARCAD1 cannot engage with the nucleosome anymore and signaling is stopped

control, or USP48$^{Iso1}$-WT- or USP48$^{Iso2}$-WT-complemented cells (Fig. 5c, d). This increased resistance was lost when USP48 depletion was accompanied either by BRCA1 or CtIP co-depletion and instead such cells became highly sensitive to camptothecin (Supplementary Fig. 9d, e). Likewise, WT-MEFs, but not $Brca1^{\Delta11/\Delta11}$ MEFs, showed a survival benefit in the presence of camptothecin following Usp48 loss (Supplementary Fig. 9f). Together these data indicate that camptothecin resistance through USP48 loss requires resection and BRCA1.

Finally, we explored the mechanisms of resistance in USP48-depleted cells. As expected, the sensitivity of both control and USP48-depleted cells to camptothecin was increased by depletion of RAD51 (Fig. 5e, f), indicating HR-GC contributes to camptothecin resistance. As the annealing of flanking repeats in SSA uses RAD52[41–43], we next assessed the contribution of SSA to camptothecin resistance by silencing RAD52

expression. Intriguingly, and in contrast to RAD51 depletion, RAD52 depletion had little impact on camptothecin sensitivity in control cells, but dramatically increased camptothecin sensitivity in USP48-depleted cells (Fig. 5e, g). Thus, a proportion of the camptothecin resistance seen on USP48 loss stems from a reliance of these cells on HR repair by SSA, which is not utilized in cells expressing USP48.

## Discussion

The data we present here establishes USP48 as a DUB that antagonizes the BRCA1 ligase function. We show that USP48 specifically counteracts BRCA1-initiated H2A ubiquitination. By limiting the extent of this ubiquitination mark USP48 affects the positioning of 53BP1 at damaged chromatin through SMARCAD1 and thus limiting resection length. We propose that the likely function of USP48 is to fine tune resection length and to avoid over-resection and mutagenic SSA. On a mechanistic level, we show that USP48 needs an auxiliary ubiquitin on the substrate to be activated, possibly allowing cross-talk between different ubiquitination sites generated by different ligases.

Methodologically, the FP assay we developed to measure site-specific deubiquitination allows rapid qualitative assessment of DUB site specificity. Quantification is complicated by variables influencing the FP signal, such as DUBs binding to free ubiquitin that changes FP properties; in addition, the ubiquitinated NCPs present a very heterogeneous substrate with a mix of mono-, di-, and tri-ubiquitinated H2A, further complicating the analysis. Nevertheless, as a qualitative measure the assay could be extended to other ubiquitinated substrates.

Under the conditions tested, most DUBs do not show site or substrate specificity, as they cleave both NCP and minimal substrate. In most cases, the minimal substrate is cleaved more efficiently, which suggests that nucleosomes are not the preferred target. With many other ubiquitinated proteins in the cell that are potentially better targets there might not be a need for clear cut substrate specificity as relative enzyme and substrate concentrations in a given situation will decide which target is cleaved. Participation in protein complexes and possible post-translational modifications will add another layer of regulation as reported for many DUBs[44].

Besides the enzymes tested there may be specific DUBs that were not included in this study, such as USP51, which has been reported to be specific for H2A$^{168ub}$ [45]. Of the DUBs we did test USP48 seems to be unique in showing clear intrinsic substrate specificity. Our biochemical analysis suggests that USP48 is specific for H2A ubiquitinated by BRCA1-BARD1.

Interestingly, USP48 reaches its full catalytic potential when more than one lysine on the BRCA1 site is ubiquitinated or when ubiquitin is located at the polycomb site at K119. A similar requirement for multiple ubiquitination events has been reported for the proteasome-associated DUB USP14[46], which cleaves supernumerary chains. However, it is not clear whether USP14 cleaves any target with multiple sites ubiquitinated or only selected substrates, a question that can be extended to USP48. Our data suggest that USP48 is fully active on H2A$^{BRCA1ub}$, but not on H2A$^{PRC1ub}$ nor H2A$^{168ub}$, showing that it indeed only acts on a selected substrate. This suggests a two-fold regulatory switch for USP48: the first being the intrinsic target specificity for the BRCA1 site on the nucleosome and the second the dependence on the auxiliary ubiquitin for full catalytic activation. Although USP14 cleaves supernumerary chains marking targets for degradation, we show that USP48 can cleave signaling monoubiquitination highlighting that a similar regulatory mechanism is employed in two profoundly different pathways. In

cells recruitment of USP48 requires BRCA1, but in vitro there is no additional affinity for ubiquitin-modified over unmodified nucleosomes, suggesting that the mechanism of accumulation may not necessarily be at the level of ubiquitin binding. It is possible that other BRCA1-mediated events encourage USP48 recruitment, but these remain to be explored.

In cells we show that resection directed by BRCA1-BARD1 can be countered by the DUB activity of USP48. Manipulation of USP48 has striking consequences for DNA end resection and cell survival following exposure to camptothecin. Without USP48 activity cells exhibit increased DNA resection, which is dependent on BRCA1 and the remodeler SMARCAD1. Moreover consistent with our previous report[10] that BRCA1 and subsequent SMARCAD1 activity acts to determine 53BP1 positioning around DNA break sites, USP48 loss results in even greater spread of 53BP1 at IRIF. Remarkably, these molecular events appear to be reflected in the increased use of both GC and SSA in USP48-depleted cells. Indeed, we show that increased camptothecin resistance seen in USP48-depleted cells is dependent on RAD51 and on RAD52, indicating use of SSA in USP48-depleted cells not present in controls. Conversely, overexpression of USP48 drastically shortens resection lengths in a manner that can be reversed by expression of a protease resistant H2A-Ub fusion. That chromatin remodeling at damaged sites is capable of promoting extended resection is supported by previous reports. For example, in yeast the SMARCAD1 ortholog Fun30, which acts to counter Rad9, can promote hyper-resection when tethered artificially to chromatin[47]. In this case, extended resection appears to be due to improved targeting of Fun30 to damaged chromatin and uncoupling it from cell cycle control[47]. Similarly, in human cells, extended resection is observed in the absence of 53BP1, or its recruitment pathway[9,48–51], and RAD52-dependent SSA becomes the dominant repair pathway[51].

Our data suggest a model in which USP48-mediated removal of the H2A$^{BRCA1ub}$ mark on chromatin restrains subsequent SMARCAD1 function, thereby halting the mobilization of 53BP1 and providing a boundary for resection (Fig. 6a). In this model the opposing activities of BRCA1-BARD1 vs. USP48 determine local H2A$^{BRCA1ub}$ modifications and consequently, in the presence of 53BP1, are capable of directing the degree of DNA end resection. We speculate that when USP48 is absent, H2A modification is sustained even at low levels of BRCA1-BARD1 accumulation so that SMARCAD1-mediated nucleosome sliding or H2A/B eviction[52] extends more widely from damaged sites than in controls.

The observation that a H2A-Ub fusion rescues USP48 overexpression phenotypes is intriguing in the light of our biochemical data indicating USP48 needs an auxiliary ubiquitin for efficient cleavage. Although it is clear that the H2A-Ub fusion requires SMARCAD1 to restore markers of HR in BRCA1-deficient cells[10], the precise requirements for SMARCAD1's reading of the H2A-Ub C terminus are not yet known. It is possible that the C-terminally linked ubiquitin fusion, due to its orientation and accessibility, is an efficient signaling entity and thus alleviates the need for an additional ubiquitin to initiate signaling on the endogenous substrate. Alternatively, the results may indicate that one ubiquitin on the BRCA1 site is enough to initiate signaling but a second, auxiliary ubiquitin is needed to switch it off, through activation of USP48. This is an intriguing model given recent data suggesting that BRCA1 activity is evolutionarily "underpowered"[53] and thereby implying that too much activity may be toxic. Such a model would suggest two distinct ubiquitination sites: a signaling site responsible for SMARCAD1 recruitment and an auxiliary site, activating USP48 to cleave the ubiquitin on the signaling site. The auxiliary ubiquitin could be conjugated to one of the two unoccupied lysines on the

BRCA1 site or may be placed by another E3 ligase on a different site, possibly by the PRC1 E3 ligase, at K118/119, providing a potential step in the observed crosstalk between DNA damage and transcriptional regulation (Fig. 6b, c)[23,54,55].

Our data supports the idea of a continuum between increased resection that promotes GC and hyper-resection leading to SSA as we see both an increase in RAD51 foci and GC repair outcomes as well as increased SSA and RAD52 dependency when USP48 is inactive or reduced. Although one model suggests ever longer ssDNA lengths may promote SSA over GC, another possibility is that a proportion of hyper-resection does not reveal direct repeat sequences either side of the break and thus encourages GC, whereas other extended track lengths expose such sequences and SSA results.

Repetitive elements are numerous in the human genome so that restricting SSA would be expected to be significant in preventing large-scale rearrangements that cause deletions of sequences located between the repeats. Thus, high levels or activity of USP48 would be expected to decrease resection lengths and thus reduce GC, and phenocopy aspects of BRCA1 loss. In contrast USP48 loss or decreased activity would be expected to increase resection lengths and favor SSA. Both outcomes might be expected to be mutagenic. Evidence that increased SSA is associated with cancer, come, e.g., from the examination of T-cell lymphoma in *Ataxia Telangiectasia Mutated* TM-deficient mice, which is suppressed following Rad52 deletion[56]. We predict that an imbalance in the BRCA1-BARD1-USP48 circuit could have deleterious consequences for genome stability and be significant in the prevention or progression of cancer.

Finally, these data also suggest a further mode of resistance against poly (adenosine diphosphate [ADP]) ribose polymerase (PARP) or topoisomerase inhibitor treatment in cancer therapy. However, in contrast to 53BP1 loss, which can restore HR repair to BRCA1-deficient tumors, USP48 loss is not an expected mechanism of tumor resistance in BRCA1 patients as hyper-resection following USP48 loss requires BRCA1 function. Nevertheless, USP48 loss of function would be expected to increase the resistance of tumors in which the BRCA1-resection pathway is intact to agents that force a reliance on HR-mediated repair.

## Methods

**Plasmids**. Flag-HA-USP48 isoform 2 was a gift from Wade Harper (Addgene plasmid 22585). RNF168 RING domain (1–113) was cloned into pETNKI-His-SUMO2-LIC-kan. Plasmids for RING1b(1–159)/BMI1(1–109) RING domain expression were described in ref. [57]. UBCH5C (UBE2D3) plasmid was a gift from P Jackson (Stanford University School of Medicine). BRCA1 and BARD1 constructs inpCOT7N vector were a gift from Rachel Klevit (University of Washington). USP15 and USP11 cDNA were a gift from Hidde Ploegh (Whitehead Institute for Biomedical Research). USP12 and UAF1 plasmids were a gift from Martin Cohn (University of Oxford). For recombinant protein expression in insect cells USP15, USP16, USP48, and BAP1 were cloned into pFastBacNKI-his-3C-LIC, USP1, USP3, and USP12 were cloned into the pFastBacHTb vector. UAF1 was cloned into pFastBac1. USP7 was cloned into pGEX6p-1 vector and expressed in *Escherichia coli*. USP11 was cloned into pET-NKI His-3C-LIC[58] vectors and expressed in *E. coli*.

**Cloning USP48 isoform 1**. USP48 isoform 1 was cloned from an isoform 2 construct by inserting the missing piece using a gene block (IDT/ sequence in Supplementary Table 2). USP48 isoform 1 was then cloned into pFastBacNKI-his-3C-LIC for expression in insect cells.

For experiments in cells, USP48 isoform 2 was cloned into pcDNA5/FRT/TO with addition of an N-terminal FLAG tag and USP48 isoform 1 cloned pcDNA5.1/FRT/TO puro(N)GFP Tobacco Etch Virus cleavage site (TEV) FLAG 3C. Point mutations to generate siResistance against both USP48-Ex5 and USP48-Ex11 siRNA sequences were generated for both isoforms and FLAG-USP48$^{Iso2}$ catalytic dead (C98S) were made by site-directed mutagenesis. All constructs were confirmed by sequencing (Source Biosciences).

All primers used for cloning and mutagenesis are given in Supplementary Table 2. HA-H2A and HA-H2A$^{KR2}$-Ub (H2A-K13,15,118,119,125,127,129R-Ub-

Kless) were described previously[10] and all H2A constructs including HA-H2A^KR1-Ub (H2A-125,127,129R-Ub-Kless) were originally synthesized by Genscript.

**Protein expression and purification**. All *E. coli* cells were grown in Lysogeny broth (LB) medium and all Sf9 insect cells in serum-free InsectXpress medium (BioWhittaker) supplemented with Penicillin/Streptomycin/Amphotericin (BioWhittaker).

**Ubiquitin**. Protein was purified from *E. coli*. Cells were grown at 37 °C until an optical density (OD) of 0.8 was reached and then induced with 400 μM Isopropyl β-D-1-thiogalactopyranoside (IPTG). Protein was expressed for h hours at 28 °C. Cells were collected in lysis buffer (50 mM TRIS pH 7.5, 150 mM NaCl, 1 mM TCEP, 2 mM Imidazole) with Complete EDTA-free protease inhibitor (Sigma) and lysed by sonication. Lysate was cleared by spinning down at 21,000 × g. Perchloric acid (2%) was slowly added to the supernatant while stirring on ice. The sample was cleared again by centrifugation at 20,000 × g and the supernatant was dialyzed against 50 mM ammonium acetate pH 4.5. The sample was then loaded on a SP HP column (GE Healthcare) and eluted with a linear salt gradient ranging from 0 to 500 mM NaCl. The sample was subsequently purified on a Superdex75 size exclusion column (GE Healthcare) in 50 mM TRIS pH 8.0 and 100 mM NaCl.

**hUBA1**. Protein was expressed in *E. coli*. Cells were grown at 37 °C until an OD of 0.8 and then induced with 200 μM IPTG. Temperature was set to 18 °C and protein was expressed overnight. Cells were collected in lysis buffer (50 mM TRIS pH 8, 100 mM NaCl, 1 mM β-mercaptoethanol) with Complete EDTA-free protease inhibitor (Sigma) and lysed by sonication. Lysate was cleared by spinning down at 21,000 × g and loaded on TALON beads (Clontech Laboratories, Inc.). Beads were washed with 20 complete volumes of lysis buffer +6 mM Imidazole and eluted in lysis buffer + 300 mM Imidazole. Sample was diluted to 50 mM NaCl and injected into a Resource Q anion exchange column (GE Healthcare). Protein was eluted using a linear salt gradient ranging from 50 mM NaCl to 600 mM NaCl. In a final step, the sample was purified on a Superdex 200 size exclusion column (GE Healthcare) in lysis buffer. Samples were concentrated and stored at − 80 °C.

**UBCH5C(UBE2D3)**. Protein was expressed in *E. coli*. Cells were grown at 37 °C until an OD of 0.8 and then induced with 200 μM IPTG. Temperature was set to 18 °C and protein was expressed overnight. Cells were collected in lysis buffer (50 mM TRIS pH 7.5, 150 mM NaCl, 1 mM TCEP, 2 mM Imidazole) with Complete EDTA-free protease inhibitor (Sigma) and lysed by sonication. Lysate was cleared by spinning down at 21,000 × g and loaded on TALON beads (Clontech Laboratories, Inc.). Beads were washed with 20 CV of lysis buffer + 6 mM Imidazole and eluted in lysis buffer + 300 mM Imidazole. His-tag was cleaved overnight at 4 °C with 3 C protease, dialyzing against gel filtration buffer (25 mM HEPES pH 8, 150 mM NaCl, 5 mM dithiothreitol (DTT)). Protease and uncleaved protein were removed using TALON beads and sample was purified by size exclusion chromatography on a Superdex 75 16/60 column (GE). Samples were concentrated and stored at − 80 °C.

**RNF168 RING domain (residues 1–113)**. Protein was expressed in *E. coli*. Cells were grown at 37 °C until an OD of 0.6 and the induced with 200 μM IPTG. Temperature was set to 18 °C and protein was expressed overnight. Cells were collected in lysis buffer (50 mM TRIS pH 8, 500 mM NaCl, 1 μM ZnCl2, 1 mM TCEP, 2 mM Imidazole) with Complete EDTA-free protease inhibitor (Sigma) and lysed by sonication. Lysate was cleared by spinning down at 21,000 × g and loaded on TALON beads (Clontech Laboratories, Inc.). Beads were washed with 20 CV of lysis buffer + 10 mM Imidazole and eluted in lysis buffer + 300 mM Imidazol. His-Sumo tag was cleaved overnight with SENP protease at 4 °C dialyzing against dialysis buffer (50 mM TRIS pH 8, 250 mM NaCl, 1 μM ZnCl2, 1 mM TCEP). Protease, uncleaved sample, and His-Sumo were removed with TALON beads. Sample was diluted to a salt concentration of 50 mM NaCl and loaded onto a Heparin column (GE). Sample was eluted with a salt gradient ranging from 50 to 1,000 mM NaCl in 12 CV. Fractions containing RNF168 were combined and further purified by size exclusion chromatography using a Superdex 75 16/60 column (GE Healthcare) in 50 mM TRIS pH 8, 150 mM NaCl, 1 μM ZnCl2, and 1 mM TCEP. Samples were concentrated and stored at − 80 °C.

**RING1b/BMI1 RING domain**. Purification of RING1b/BMI1 RING domain constructs was done as described before[57] with slight changes. *E. coli* cells were grown at 37 °C until an OD of 0.8 was reached and then induced using 200 μM IPTG. Protein was expressed over night at 18 °C. Cells were collected in the morning in lysis buffer (50 mM TRIS pH 7.5, 150 mM NaCl, 2 μM ZnCl2, 1 mM DTT) and lysed by sonication. Lysate was cleared by spinning at 21,000 × g and supernatant was loaded on glutathione sepharose beads (GE Healthcare). The beads were washed with 200 ml lysis buffer and protein was eluted using 20 ml lysis buffer + 20 mM glutathione. The glutathione S-transferase (GST) tag was cleaved overnight using 3C protease at 4 °C. The cleaved sample was purified on a Superdex 75 16/60 size exclusion column in lysis buffer with a GST-trap fitted at the end to remove uncleaved protein and free GST. Samples were concentrated and stored at − 80 °C.

**BRCA1-BARD1 RING domain**. BRCA1 (1–303) and BARD1 (1–306) were co-expressed in *E. coli*. Cells were grown in LB at 37 °C until OD of 0.6 was reached and then induced with 100 μM IPTG. Protein was expressed for 4 h at 37 °C. Cells were collected in lysis buffer (25 mM HEPES pH 7.5, 150 mM NaCl, 1 mM TCEP, 5 mM Imidazole) and lysed by sonication. Lysate was cleared by spinning at 21,000 × g and supernatant was loaded onto chelating sepharose beads (GE healthcare) charged with Ni^{2+} in gravity flow columns. Beads were washed with 20 CV lysis buffer + 30 mM Imidazole. Protein was eluted in lysis buffer + 300 mM Imidazole. Salt concentration was diluted to 50 mM NaCl and sample was loaded on a Resource S cation exchange column (GE Healthcare). Protein was eluted with a salt gradient (50–1,000 mM NaCl, 25 mM HEPES pH 7.5, 1 mM TCEP) and fractions containing BRCA1-BARD1 were pooled. Pooled fractions were further purified by size exclusion chromatography on a Superdex 75 column (GE Healthcare). Samples were concentrated and stored at − 80 °C.

**USP1/UAF1complex**. Proteins were expressed and purified as described in ref. [59]. Proteins were coexpressed by Baculovirus expression in Sf9 insect cells. Cells were infected at a density of $1 \times 10^6$ cells per ml and grown for 72 h. Cells were collected in lysis buffer (50 mM TRIS pH 8, 150 mM NaCl, 2 mM TCEP) and complete EDTA-free protease inhibitor (Sigma) was added. Cells were lysed by sonication and sample was cleared by centrifugation at 21,000 × g for 30 min. Supernatant was loaded on His-affinity column (GE Healthcare) and column was washed with lysis buffer + 50 mM Imidazol. Sample was eluted with 500 mM Imidazol. The eluted sample was then loaded on a Strep-affinity column (IBA Life Science) and eluted using lysis buffer + 2.5 mM desthiobiotin. The sample was then purified on a Superdex 200 size exclusion column (GE Healthcare) in lysis buffer. Samples were concentrated and stored at − 80 °C.

**USP7**. Protein was expressed in BL21(DE2)Rosetta2 cells. Cells were grown at 37 °C in Terrific Broth until an OD of 1.8–2.0 and induced with 100 μM IPTG. Temperature was set to 18 °C and protein was expressed overnight. Cells were lysed using Emulsiflex in lysis buffer (50 mM HEPES pH 7.5, 250 mM NaCl, 1 mM EDTA, 1 mM DTT) + 0.1 mM phenylmethylsulfonyl fluoride and 1 mg DNAse1. Lysate was cleared by centrifugation at 20,000 × g and supernatant was loaded on Glutathione Sepharose 4B beads (GE Healthcare) in gravity flow column. Beads were washed in lysis buffer and eluted in lysis buffer + 15 mM reduced glutathione. GST-tag was cleaved overnight with 3C protease dialyzing against 10 mM HEPES pH 7.5, 50 mM NaCl, 1 mM EDTA, 1 mM DTT. Sample was purified using a ResourceQ anion exchange column (GE Healthcare) using a salt gradient (50–500 mM NaCl, 10 mM HEPES pH 7.5, 1 mM DTT). USP7 containing fractions were pooled and further purified by size exclusion chromatography on a Superdex 200 column (GE Healthcare). Samples were concentrated and stored at − 80 °C.Please spell out TB in text, as it is mentioned only once.in text

**USP11**. Protein was expressed in *E. coli* and purified as described before[60]. USP11 was expressed in BL21(DE3) *E. coli* cells. Cells were grown in LB at 37 °C until an OD of 0.8 was reached and were then induced with 200 μM IPTG. Protein was expressed overnight at 20 °C. Cells were collected in lysis buffer (20 mM TRIS pH 7.5, 150 mM NaCl, 5 mM β-mercaptoethanol) and lysed by sonication. Sample was cleared by centrifugation at 21,000 × g for 30 min and the supernatant was loaded on chelating sepharose beads charged with Ni^{2+}. Beads were washed with lysis buffer and sample was eluted in lysis buffer + 350 mM Imidazol.

**USP12**. Same purification as for USP1/UAF1 until the elution from the His-affinity column. For USP12 the His-tag was then cleaved overnight using TEV protease, while dialyzing against lysis buffer overnight. After cleavage, the sample was flown over a His-affinity column again to remove uncleaved protein and protease (sample was collected from the flow through). The sample was then purified on a Superdex 200 size exclusion column (GE Healthcare) in lysis buffer. Samples were concentrated and stored at − 80 °C.

**USP15, USP16, and USP48 (isoform1 and 2)**. Proteins were expressed in Sf9 insect cells for 48–72 h (until viability dropped below 90 %). Cells were lysed by sonication in 25 mM HEPES pH 7.5, 300 mM NaCl, 2 mM DTT, and 5 mM Imidazole supplemented with Pierce EDTA-free protease inhibitor (Thermo Fisher). Lysate was cleared by centrifugation at 21,000 × g at 4 °C and the supernatant was loaded on chelating sepharose beads (GE) charged with Ni^{2+}. Beads were washed with 20 column volume of lysis buffer + 30 mM Imidazole and eluted in lysis buffer + 300 mM Imidazole. The His-tag was cleaved using His-tagged 3C protease overnight at 4 °C, while dialyzing against 25 mM HEPES 7.5, 150 mM NaCl, 1 mM DTT. The sample was then run over chelating sepharose beads charged with Ni^{2+} to remove protease and uncleaved sample, and subsequently purified by size exclusion chromatography using a Superdex S200 16/60 column (GE) in 25 mM HEPES 7.5, 150 mM NaCl, 1 mM DTT. Samples were concentrated and stored at − 80 °C.

**USP3**. Same as USP48 but with 500 mM NaCl and 1 μM ZnCl$_2$ in the lysis buffer and TEV protease was used to cleave the tag. Samples were concentrated and stored at − 80 °C.

**BAP1**. The protein was expressed in SF9 insect cells using the Baculovirus expression system as described in ref. [61]. Cells were infected for 72 h and collected in lysis buffer (50 mM TRIS pH 8.0, 200 mM NaCl, 50 mM Imidazol, 0.5 mM TCEP) supplemented with complete EDTA-free protease inhibitor (GE Healthcare) and lysed by sonication. The lysate was cleared by centrifugation at $20,000 \times g$ and the supernatant was loaded on chelating sepharose beads (GE Healthcare) charged with Ni$^{2+}$. Beads were washed in 15 column volumes wash buffer and eluted in elution buffer (20 mM TRIS pH 8.0, 150 mM NaCl, 500 mM Imidazole pH 8.0, 10% Glycerol, 0.5 mM TCEP). The sample was then purified on an anion exchange column (Poros HQ, Applied Biosystems). The sample was applied in 20 mM TRIS pH 8.0, 50 mM NaCl, 5% Glycerol, 0.5 mM TECP, and eluted using a linear salt gradient ranging from 50 to 750 mM NaCl. The sample was then further purified on a Superrose 6 size exclusion column (GE Healthcare) in 10 mM HEPES pH 7.5, 150 mM NaCl, 10 % Glycerol, 0.5 mM TCEP. Samples were concentrated and stored at − 80 °C.

**ASXL1 (1–390)**. Protein was expressed and purified as described in ref. [61]. The protein was expressed in *E. coli*. Cells were grown at 37 °C until an OD of 0.6 was reached and then induced with 500 μM IPTG. The protein was expressed at 25 °C for 4–6 h. The cells were collected in lysis buffer (50 mM Tris pH 8.0, 200 mM NaCl, 50 mM Imidazole pH 8.0, 0.5 mM TCEP) supplemented with complete EDTA-free protease inhibitor (GE Healthcare) and lysed by sonication. The lysate was cleared by centrifugation at $20,000 \times g$ and the supernatant was loaded on chelating sepharose beads (GE Healthcare) charged with Ni$^{2+}$. The beads were washed in 10 column volumes of lysis buffer and the sample was eluted in 20 mM Tris pH 8.0, 100 mM NaCl, 500 mM Imidazole pH 8.0, 10% Glycerol, and 0.5 mM TCEP. The sample was then purified by cation exchange chromatography on a Poros HS column (Applied Biosystems). The sample was applied in 20 mM Bis-Tris pH 6.5, 50 mM NaCl, 5% Glycerol, 0.5 mM TCEP, and eluted with a linear salt gradient ranging from 50 to 750 mM NaCl. In a final step the sample was purified by size exclusion chromatography on a Superdex 75 column (GE Healthcare) in 10 mM HEPES pH 7.5, 150 mM NaCl, 10 % Glycerol, 0.5 mM TCEP.

**Nucleosome reconstitution**. Histones and 147 bp DNA with Widom601 strong positioning sequence were purified, octamers folded, and nucleosomes reconstituted by salt dialysis as described previously[62,63]. To assemble the Octamer purified recombinant histones were combined at equimolar concentrations (20 μM) and dialyzed against three changes of 2 l of refolding buffer (20 mM TRIS pH 7.5, 2 M KCl, 5 mM β-mercaptoethanol, 1 mM EDTA). Octamers were then purified by gel filtration on a Superdex 200 column (GE Healthcare) in refolding buffer. NCPs assembled by combining Octamer and DNA at a ratio of 1 : 0.9 (using 7 μM Octamer), adjusting the salt concentration so it is kept at 2 M KCl at all times. The samples were then dialyzed against 10 mM TRIS pH 7.5, 2 M KCl, 1 mM EDTA, 1 mM DTT at 4 °C. The salt concentration was then lowered gradually to 250 mM KCl over a period of 18 h. Finally, the sample was dialyzed against 20 mM TRIS pH 7.5, 150 mM NaCl, 1 mM DTT for 4 h (or overnight) at 4 °C.

**Cell lines**. Flp-In Doxycyclin-inducible HeLa parent cell lines were a gift from Grant Stewart, University of Birmingham, and were clonally selected in the Morris lab for Tet-Inducible expression. FlpIn HeLa cell lines were grown in DMEM (Sigma), 10 % Tetracycline-Free Fetal Calf serum (Clontech) supplemented with 1 % penicillin/streptomycin. All other cell lines were grown in DMEM supplemented with 10% fetal calf serum (Sigma) and 1 % penicillin/streptomycin. Mycoplasma testing was by Hoechst DNA staining. Stable doxycycline-inducible cell lines were made by co-transfection of pcDNA5/FRT/TO-FlagUSP48$^{Iso2}$-siResistant-WT or -C98S constructs with pOG44 recombinase into HeLa-FlpIn cells. Clones were selected in hygromycin (400 μg ml$^{−1}$) and expanded. Flag-USP48 expression following 48–72 h induction with doxycycline (1 μg ml$^{−1}$) was confirmed by western blotting. HeLa-FlpIn EGFP-USP48$^{Iso1}$ cells were made likewise in the Sixma lab. *Brca1*$^{Δ11/Δ11}$ MEFs were a kind gift from Andre Nussenzweig (National Cancer Institute National Institutes of Health, Rockville USA). WT MEFs were generated in the Morris lab. U2OS-GFP reporter cell lines for GC (DR3), NHEJ (EJ5), and SSA were a kind gift from Jeremy Stark (City of Hope, Duarte, CA).

**Transfections**. siRNA transfections were performed using Dharmafect1 (Dharmacon) and DNA plasmids using FuGENE 6 (3 μl:1 μg FuGENE:DNA) (Promega) following the manufacturer's protocols. Cells were grown for 48 h post transfection before treatment and collection. With the exception of Supplementary Fig. 5a, all siRNA against USP48 was a combination of USP48-Ex5 and USP48-Ex11 siRNA sequences. All siRNA sequences are given in Supplementary Table 3.

**Western blottings**. A full list of antibodies used for western blottings can be found in Supplementary Table 4. Uncropped western blottings are shown in Supplementary Fig. 10.

**Kinetics on minimal substrate Ubiquitin-Rhodamine**. Ubiquitin linked to a cleavable small peptide labeled with Rhodamine (Ub$^{RHO}$, UbiQ) was used as a substrate. Reaction was followed through increase of fluorescence intensity at 590 nm due to liberation of free fluorescent Rhodamine from the quenched substrate upon cleavage by a DUB. The assays were done in 384-well plates (Corning, flat bottom, low flange) in a Pherastar plate reader (BMG Labtech) at 30 °C using an assay buffer of 25 mM HEPES pH 7.5, 150 mM NaCl, 5 mM DTT, and 0.05 % TWEEN-20. For full Michaelis–Menten analysis substrate was titrated starting from 30 μM in eight two-fold dilutions, whereas enzyme concentration was kept constant. Enzyme concentrations are indicated in Supplementary Fig. 1. Cleavage was started by addition of the respective DUB. The initial velocity of the reaction was calculated from the slope of the linear phase of the curve and was plotted against the substrate concentration and fitted to the Michaelis-Menten equation using the program GraphPad Prism.

For single point assays (Supplementary Fig. 1e), 2 μM of substrate and different concentrations of enzyme were used. Assays were done at 30 °C. The different enzyme concentrations are indicated in the figure.

**Fluorescent labeling of ubiquitin**. Ubiquitin with a cysteine residue introduced at the N terminus right after the methionine at position 1 was labeled using maleimide-linked TAMRA dye. Ubiquitin (250 μM) was labeled with 1500 μM TAMRA (5)-Maleimide (Setareh) at 4 °C overnight. Excess dye was removed by size exclusion chromatography using a Superdex 75 16/60 column (GE) in 50 mM TRIS pH7.5, 150 mM NaCl, and 1 mM DTT.

**Ubiquitination assay**. Nucleosomes were ubiquitinated using 0.5 μM hUBA1, 1 μM UbcH5C (UBE2D3), 1 μM E3 RING domain (RNF168, RING1B/BMI1, or BRCA1-BARD1), 5 μM NCP, 20 μM ubiquitin or $^{TAMRA}$Ub, and 3 mM ATP in 25 mM HEPES pH 7.5, 150 mM NaCl, 3 mM MgCl$_2$, and 1 mM DTT for 60 min at 30 °C. Substrates for Fig. 1c were ubiquitinated for 40 min, all the others for 60 min (as stated). Ubiquitinated NCPs were then gel filtered on a Superose 6 Increase column using the Akta micro purifier system (GE) in 25 mM HEPES pH 7.5, 150 mM NaCl, 1 mM DTT.

**Double ubiquitination at polycomb and BRCA1 site**. NCPs containing the H2A K118R were used to assure only K119 is ubiquitinated by RING1B/BMI1. Mutant NCP were ubiquitinated with RING1B/BMI1$^{RING}$ and gel filtered on the Superose 6 increase using the Akta micro purifier system (GE Healthcare). Purified NCP ubiquitinated at K119 were then ubiquitinated a second time using BRCA1-BARD1$^{RING}$ and $^{TAMRA}$Ub, generating NCP ubiquitinated on K119 with unlabeled ubiquitin and on the BRCA1 site with $^{TAMRA}$Ub.

**FP assay to measure DUB activity**. FP assays were done in 384-well plates (Corning, flat bottom, low flange) in a Pherastar plate reader (BMG Labtech) at 30 °C using an assay buffer of 25 mM HEPES pH 7.5, 150 mM NaCl, 5 mM DTT, and 0.05 % TWEEN-20 (Sigma). NCPs ubiquitinated with $^{TAMRA}$Ub on either of the three sites were used at 2 μM label concentration. A quantity of 500 nM of the respective DUB was added to start the assay and reaction progression was followed by measuring FP signal at Excitation: 540 nm, Emission: 590/590 nm. A reaction without DUB was run in parallel to subtract baseline drift from the experimental data. An exponential function $FP = (FP_0 − FP_{min}) \times e^{−k_{obs} \times t} + FP_{min}$, where FP$_0$—FP is the signal of ubiquitinated substrate and FP$_{min}$—FP the signal of free $^{TAMRA}$Ub, was fitted to determine $k_{obs(NCP)}$ as a measure of the enzymes processivity. To normalize the data to the intrinsic DUB activity on minimal substrate cleavage of 2 μM Ub$^{Rho}$ under the same conditions. The curve was fitted to obtain $k_{obs(Rho)}$. $k_{obs(NCP)}$ was then normalized to $k_{obs(Rho)}$ using the equation $k_{obs(norm)} \frac{k_{obs(NCP)} * [E_{Rho}]}{k_{obs(Rho)} * [E_{NCP}]}$, where $[E_{Rho}]$ is the concentration of the DUB in the reaction with minimal substrate and $[E_{NCP}]$ is the concentration of the DUB in the reaction with NCP.

**Biotinylation and USP48 inactivation and SPR measurements**. Measurements were done on a Biacore T200 (GE). NCPs of different ubiquitination status at a concentration of 2 μM were biotinylated over night at 4 °C using EZ-link Sulfo-NHS-LC-Biotin (Thermo Fisher) with a 2:1 excess of NCP over biotin. Biotinylation was stopped by adding 1 mM TRIS pH 7.5. USP48 at 30 μM was inactivated by incubating with 25 mM Iodoacetamide for 20 min at room temperature. Excess Iodoacetamide was removed using Zeba Spin Desalting Columns (Thermo Fisher) according to manufacturer's protocol. The buffer used for SPR experiments was 25 mM HEPES pH 7.5, 150 mM NaCl, 0.05% TWEEN-20, 1 mg ml$^{−1}$ bovine serum albumin (BSA), 1 mg ml$^{−1}$ Dextran. Biotinylated NCPs were immobilized on an SA chip (Series S SA chip, GE) to roughly 100 RU and inactivated USP48 was flown over at varying concentrations with two-fold dilutions starting at 2.56 μM. A reference flow cell without immobilized NCP was included to subtract unspecific binding. Data were processed using GraphPad Prism 7.0.

**NCP cleavage reactions with USP48**. All cleavage reactions were done in reaction buffer 25 mM HEPES pH 7.5, 150 mM NaCl, 5 mM DTT. NCPs ubiquitinated on either one of the H2A ubiquitination sites with $^{TAMRA}$Ub were used and

reaction was started by addition of USP48. Concentrations are indicated in the figures. Samples were taken at the indicated time points and reaction was stopped by addition of SDS-loading dye. Samples were separated on a NuPage 4–12 % Bis-TRIS SDS gel in MES buffer (Thermo Fisher) and the fluorescence signal was read out on a Typhoon FLA-9500 gel scanner (GE Healthcare). Quantification of individual bands was achieved by measuring the fluorescence intensity and relating it to the total intensity of each lane at a known concentration of dye used in the respective assay. This way each band represents a fraction of the concentration of dye used in the experiment. To convert to molar concentration of ubiquitinated histones the dye concentration was divided by the number of labeled ubiquitins present on the respective histone species (H2Aub3, H2Aub2, and H2Aub1).

**USP48 kinetic modeling**. USP48 cleavage reactions as described above were done at varying substrate and enzyme concentration to allow subsequent kinetic modeling. The different conditions are indicated in Supplementary Figs. 3–4. The concentrations of the different reaction species were quantified according to the fluorescence signal to yield molar concentrations of the fluorophore. The molar concentration of fluorophore was then dividied by the number of fluorophores present on each individual reaction species (three for H2Aub3, two for H2Aub2 and 1 for H2Aub and ub) to determine the molar amounts of each species. The quantified data were used to globally fit the three different catalytic rates defined in Fig. 2e, $k_{cat(ub3)}$, $k_{cat(ub2)}$, and $k_{cat(ub1)}$. The association constant was fixed to $k_{on} = 20 \, \mu M^{-1} \, s^{-1}$, an order of magnitude we routinely observe in SPR experiments. $k_{off}$ was determined by an initial round of fitting where it was allowed to vary between 2 and 20 $s^{-1}$ to reflect the range of the experimentally determined Kd between 100 nM as approximated by SPR and 1 μM as estimated by gel shift assays. The model consistently chose 20 $s^{-1}$ as the best fit value and $k_{off}$ was fixed to this value in subsequent fitting steps. All fitting was done using the software KinTek Explorer Version 6.3.170707. The amount of different conditions needed for a reliable fit of the variables has been determined in an iterative fashion by adding more conditions and subsequently analyze the goodness of fit (as described in statistics, see below). With the number of conditions used now $\chi^2$ values fall into a well-defined minimum, suggesting that the fitted parameters are well restrained by the experimental data.

**DNA repair reporter assays**. U20S-DR3-GFP (GC), U20S-SA-GFP (SSA), and U20S-EJ5-GFP (NHEJ) were a generous gift from Jeremy Stark (City of Hope, Duarte USA). U20S reporter cell lines were simultaneously co-transfected with siRNA using Dharmafect1 (Dharmacon) and DNA (RFP, or Flag-USP48 and I-Sce1 endonuclease expression constructs) using FuGene6 (Promega) respectively. After 16 h the media was replaced and cells were grown for a further 48 h before fixation in 2% paraformaldehyde (PFA). RFP and GFP double positive cells were scored by FACS analysis using a CyAn flow cytometer and a minimum of 10,000 cells counted. Data were analyzed using Summit 4.3 software. Each individual experiment contained three technical repeats and normalized to siRNA controls or to WT-complemented cells. Graphs shown are combined data from a minimum of three independent experiments and error bars show SE.

**Colony assays**. Cells were plated at $2 \times 10^5$ cells per ml in a 24-well plate and treated as indicated in the figure legends. Cells were then trypsinized and plated at limiting dilution to form colonies and grown on for 10–14 days. Colonies were stained using 0.5% crystal violet (BDH Chemicals) in 50% methanol and counted. Each individual experiment contained three technical repeats and is normalized to untreated controls. Graphs shown are combined data from a minimum of three independent experiments and error bars show SE.

**Laser micro-irradiation**. Laser micro-IR experiments were performed on BrdU-presensitized cells (10 μM BrdU, 24 h) as described[64] using a Zeiss PALM MicroBeam equipped with a 355 nm UV-A pulsed-laser and the × 40 objective with laser output at 40%, assisted by the PALMRobo-Software supplied by the manufacturer. Cells were fixed after 30 min in 4 % PFA before staining by immunofluorescence.

**Measurement of resection tracks (BrdU)**. Twenty-four hours before fixation cells were incubated with 10 μM BrdU and then 10 μM Olaparib for the last 16 h of treatment. Cells were trypsinized and resuspended in ice-cold phosphate-buffered saline (PBS) to a concentration of $10 \times 10^5$ cells per ml. In order to lyse the cells, 2 μl of sample was placed on a slide and mixed with 7 μl of spreading buffer (200 mM Tris pH 7.4, 50 mM EDTA, 0.5% SDS) and incubated for 2 min. Slides were then placed at a shallow angle to cause the droplet to gradually run down the slide, ensuring constant movement of the droplet. Slides were fixed in 3 : 1 Methanol: Acetic Acid for 10 min and then stored at 4 °C. Slides were washed in PBS and Blocking solution (2 g BSA, 200 μl Tween-20, 200 ml PBS) and then incubated with mouse anti BrdU primary antibody. Slides were washed in PBS before being incubated with AlexaFluor donkey anti mouse 488. Images were taken on the Leica DM6000B microscope and analysis performed using ImageJ software. Lengths were calculated using a scale bar to convert pixels to μm and this ratio, of 3.493 pixels per μm, was used to measure BrdU track lengths. Fibres (> 100) per treatment were measured and plotted on a Whisker plot using Graphpad.

**Immunofluorescence**. HeLa FlpIn, HeLa-Flag-USP48[Iso2], HeLa-Flag-USP48-C98S[Iso2], or HeLa-EGFP-USP48[Iso1] cells were seeded onto coverslips and transfected with siRNA, expression constructs and/ or induced with doxcycline as described. Cells were labeled with 10 μM EdU 10 min before IR using a Gamma-cell 1000 Elite irradiator (caesium-137 source). At 2 h post IR, cells were washed briefly in CSK buffer (100 mM sodium chloride, 300 mM sucrose, 3 mM magnesium chloride, 10 mM PIPES pH 6.8) before fixation with 4% PFA for 10 min. For immunofluorescence staining, cells were permeabilized with 0.2% Triton X-100 in PBS for 10 min before blocking in 10 % fetal bovine serum in PBS. EdU was visualized by Click-iT chemistry according to the manufacturer's protocols (Life Technologies) with Alexa-647-azide. Cells were incubated with primary antibody for 1 h, washed three times in PBS, and incubated with secondary AlexaFluor antibodies for 1 h. The DNA was stained using Hoechst at 1 : 20,000.

**Microscopy**. For 53BP1/BRCA1 foci spread analysis: Images of immuno-fluorescent staining were captured on the Zeiss 510 Meta confocal microscope, using three lasers to give excitation at 647, 555, and 488 nM wavelengths. Images at each wavelength were collected sequentially at a resolution of approximately 1,024 × 1,024 pixels, using the Plan-Apochromat × 100/1.4 Oil objective. All other immunofluorescent staining was imaged using the Leica DM6000B microscope with a HBO lamp 100 W mercury short arc UV bulb light source and four filter cubes, A4, L5, N3, and Y5, to produce excitations at wavelengths 360, 488, 555, and 647 nm, respectively. Images were captured at each wavelength sequentially using the Plan Apochromat HCX × 100/1.4 Oil objective at a resolution of 1,392 × 1,040 pixels.

**Statistics**. BrdU resection tracks that were analysed by one-sided Mann–Whitney test with center values given as median. All other statistical analysis was by two-sided Student's T-test throughout. * $p < 0.05$, **$p < 0.01$, and ***$P < 0.005$. All centre values are given as the mean and all error bars are SEM. For kinetic modeling, data were fitted globally to the defined model by numerical integration and best-fit parameters were determined by finding minimum $\chi^2$ values using KinTek Explorer. The quality of the fit was evaluated by calculating the FitSpace shown in Fig. 2e.

**Data availability**. The data sets generated during the current study are available from the corresponding authors on reasonable request.

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

## Acknowledgements

Grant funding for this project was as follows. NWO-CW TOP 714.012.001, ERC 249997, Gravity CGC.nl, CRUK: C8820/A19062 (R.M.D.). J.R.M. is HEFCE funded. We thank J. Stark (City of Hope) for U20S-DR3, U20A-SA, and U20S-EJ5 cells, and I-SCE1 plasmid, Andre Nussenzweig (NCI, Bethesda, USA) for the *Brca1* $^{\Delta11/\Delta11}$ MEFs, and Fena Ochs and Claudia Lukas (University of Copenhagen) for RAD52 antibody. Alex Garvin designed HA-H2A$^{KR1}$-Ub and HA-H2A$^{KR2}$-Ub. In addition, we thank the TechHub Facility at the University of Birmingham for Microscope and FACS support.

## Author contributions

M.U. devised the screen and kinetic analysis and performed all in vitro assays. H.H.K.W. reconstituted NCPs, A.F. supervised kinetic analysis, and R.M.D. performed cell and biochemical experiments, designed experiments, and interpreted data. T.K.S. initiated the project and supervised M.U. J.R.M. contributed to data interpretation, directed the

project, and supervised R.M.D. M.U., R.M.D., T.K.S., and J.R.M. contributed to writing the paper.

## Additional information

**Competing interests:** The authors declare no competing financial interests.

