## [Peer Review File · Nature Communications]

Reviewers' Comments:

Reviewer #1:

Remarks to the Author:

BRCA1/BARD1 dimer have multiple roles in DNA double strand breaks repair. Some of them rely on its ubiquitin-ligase activity and in ubiquitination of specific targets. One of them is the ubiquitination of H2A, that is required to overcome the chromatin barrier during DNA resection. In this manuscript, the authors identify USP48 as a specific deubiquitinating enzyme of H2A, specifically the BRCA1-mediated ubiquitination, using an elegant in vitro approach. This deubiquitination works better when multiple lysines are conjugated, suggesting that the presence of one Ub stimulates the reaction.

In vivo, USP48 is recruited to DNA damage, and its depletion or lack of activity stimulates longer resection and RAD51 accumulation in BRCA1 and SMARCD1 dependent fashion. Mechanistically, affect the relocalization of 53BP1 within IRIF. As a consequence, lack of USP48 activity stimulates recombination and survival upon CPT treatment.

The experiments are well executed, and the conclusions are well supported by the experimental data. Overall, the story is well presented and interesting enough for Nat Commun audience. However, in my opinion, there are a few experiments that will strengthen the final model.

1. No RPA foci are shown in the paper, but a measurement of resection length. Is this because USP48 depletion does not affect RPA foci formation (i.e. resection initiation)? This will fit the model, so RPA foci should be included as a supplementary figure.
2. The in vitro study clearly shows the specificity of USP48 for H2A ubiquitination by BRCA1. The in vivo data support the idea as USP48 depletion is irrelevant for resection and Rad51 foci after BRCA1 downregulation. In that regard, it will strengthen the model if the hyper-recombination phenotype and hyper-resistance to CPT are also tested upon BRCA1 depletion. Along the same lines, one or both these phenotypes should be tested in the absence of CtIP, to demonstrate that rely on hyper-resection.
3. In that regard, it is difficult to conclude from the data the contribution of H2A deubiquitination in the in vivo phenotypes. It is possible that USP48 deubiquitinates other BRCA1 substrates that affect resection, recombination and cell survival. I suggest to test some additional phenotypes in cells expressing an H2A-Ub and overexpressing USP48
4. Finally, in the model, the authors propose that USP48 limits resection by counteracting BRCA1 ubiquitination activity. Is this limitation a temporal or spatial? If is a temporal limitation, USP48 should be recruited later than BRCA1 to laser lines, something the authors can easily test using fluorescently tagged BRCA1 and USP48. If is a spatial restriction, the prediction is that it will show a similar accumulation pattern as 53BP1, measured in figure 4e. This will complete the model.

Minor points:

The TAMRA gels look a bit fuzzy always in my copy of the manuscript. I am not sure is due to the conversion of the files, but it should be improved.
Supplementary figure S2a, the gels look fuzzy.

Reviewer #2:

Remarks to the Author:

"USP48 restrains resection by site specific cleavage of the BRCA1 ubiquitin mark from H2A." by Michael Uckelmann, Ruth M. Densham, Herrie H. K. Winterwerp, Alexander Fish, Titia K. Sixma, and Joanna R. Morris

This is an elegant study from the Sixma and Morris labs identifying the deubiquitinase USP48 as a regulator of H2A ubiquitination and thereby DNA end resection and DSB repair. The first part of

the paper is biochemical – here, the authors identify USP48 as an H2A DUB with site-specificity for the BRCA1/BARD1 target sites on H2A and show an interesting mechanism of USP48 control by “auxiliary” ubiquitin-modifications. The second part of the paper is cell biological – here, the authors mainly characterize the phenotypes of an USP48 knock down. These phenotypes indicate convincingly that a lack of USP48 leads to hyper-resection and consistently to hyper-recombination.

Overall, this is a very high quality study to a biological phenomenon (DSB repair pathway choice), which is extremely important but still incompletely understood. As such this paper should be very interesting to the broad readership of Nature Communications. I only have very few points that should be addressed prior to publication.

Major points:

1. Both parts of the paper are themselves convincing, but it would indeed be very nice to see a connection. I.e. does USP48 knock-down lead to enhanced levels of H2AK125,127,129ubi in damaged cells?
2. Fig. 1b: The authors show values normalized to ubiquitin-rhodamine, probably because the relative activity of the different DUB preps are different. Nonetheless, it would be desirable to also show the non-normalized data.
3. Fig. 3a: The USP48 localization to laser-induced foci is intriguing, but the authors should look at this localization in a time course experiment to establish whether USP48 localizes to DSB sites early or late. Moreover, the finding of apparently endogenous USP48 foci is also intriguing and it would be good to know under which conditions (e.g. cell cycle state) USP48 forms foci in unchallenged cells.
4. Fig. 4d: The experiment with the H2A-ubi fusion seems complicated to interpret, perhaps because the authors are using not only a covalent fusion between H2A and ubi, the constructs also lack the lysines, which are normally targeted (as far as I can extract from the methods part). I am also puzzled why this fusion on the one hand counteracts the effect of USP48 overexpression, but does not enhance DNA end resection by itself. The authors should investigate this in more detail with a fusion containing the endogenous sites on H2A and another control mutant lacking the H2A lysines (is the effect of this mutant not best explained, if it counteracted both anti- and pro-resection factors?) or remove this part from the paper.
5. Fig. 3-4: The authors use number of Rad51 foci as measure of DNA end resection and HR. Have they thought of using foci intensity as additional measure of resection.

Minor points:

1. Fig. 5c: This experiment indicates that hyperresection may lead to camptothecin resistance, which is intriguing. The Pfander lab has recently reported that in yeast hyperresection leads to camptothecin resistance (Bantele et al. 2017, eLIFE). I would encourage the authors to discuss this study, as both conditions apparently hyperactivate resection via the SMARCAD1/Fun30 pathway.

2. Even though the writing is very clear throughout the paper, there are a few sentences/expressions that are somewhat unclear and should be changed.

abstract & end of p2, l69: “auxiliary ubiquitin” – this is only defined later in the paper and remains unclear at first

abstract: “extended 53bp1 positioning”

p2, I52: "synthetic viability"

p5, I165: "track-length of ... were spread an measured"

p5, I178: "Resection promoted by the BRCA1-BARD1 E3 Ub ligase promotes SMARCAD1-mediated remodelling of chromatin-associated 53BP1."

3: Fig. 2a: The stacking of the westerns looks strange to me. If the authors want to show this data they should find a different layout.

4. Fig. 4b: The SMARCAD1 western for the siRNA control does not look very convincing. It would be good to have a better (and perhaps quantified) western. If not possible please provide an mRNA-based expression control.

Reviewer #3:

Remarks to the Author:

While the impact of BRCA1 in orchestrating DNA damage repair and tumor susceptibility is highly recognized, the role of BRCA1/BARD1 acting as ubiquitin protein ligase has been controversial (Ludwig T. et al. *Science* vol 334, 2011 and Jonkers J. et al. *Cancer Cell* 20, 2011). Previously report described the ubiquitin ligase activity for BRCA1/BRAD1 (Wu L. et al. *Nat. Genet.* 14, 1996 and Hashizume R. et al. *JBC* 276, 2001). It was thought that H2A is one of the major targets of BRCA1/BARD1 in response to DNA damage through K125/127/129 on H2A. Moreover, the author revealed that SMARCAD1 is a cascade down-stream of H2A ubiquitination to 53BP1 positioning and resection. In this manuscript, the authors deciphered the role USP48 to be a deubiquitinating enzyme that counteracts the BRCA1/BARD1-catalyzed H2A ubiquitination and further linked the function of USP48 to regulating DNA end resection and RAD51 recruitment. While addition of USP48 to the paradigm of BRCA1/BARD1-H2A-DNA end resection/RAD51 recruitment is new, the data present at this time is not enough to prove the hypothesis. Following are concerns:

1. The USP3 has the same activity as USP48 on deubiquitinating of H2ABRCA1Ub. Therefore, whether USP3 has a compensative effect on H2ABRCA1Ub modification, gene conversion and single strand annealing when USP48 is knockdown? Which one is the major regulator of H2ABRCA1Ub deubiquitination.
2. The approach bridging USP48 to H2A ubiquitination sounds candidate-driven manner.
3. The authors need to present whether USP48 promotes genome stability depend on BRCA1-H2A or not in BRCA1 mutant or loss cells.
4. The author showed USP48 loss resulting in increased end resection and RAD51 foci numbers. Whether modulating of USP48 expression affects the numbers, size, and intensity of single-strand DNA binding protein RPA loading on the resected ssDNA?
5. The colony survival assay has only been investigated in Hela cells. Whether modulating of USP48 expression affect colony survival in BRCA1 mutant or loss cells? Whether the effect of USP48 expression on cell survival is depend on counter-act on BRCA1-H2A ubiquitination cascade?
6. This manuscript chooses the list of DUBs in the DNA damage response (USP1/UAF1, USP11, USP7, USP15, USP12/UAF1) from *Nat Cell Biol.* 2014 Oct; 16(10): 1016–8. However, USP48 is not in the list of DUBs which have the features that localization of GFP–DUBs to laser micro-irradiation sites and effects on DSB repair using siRNA pools (neutral comet assay), but in the list of DUBs which G2/M affect G2/M checkpoint.
7. In figure S4d, knockdown 53BP1 seems to promote Rad51 foci formation. This is a conflict with previous studies (*Nucleic Acids Res.* 2013 Nov; 41(21): 9719-31; *Nat Struct Mol Biol.* 2010 Jun; 17(6): 688-95.).
8. The overexpressed FLAG-USP48 protein is about several folds of the endogenous USP48 (fig 3b, 5c), but in fig 3e, FLAG-USP48 (WT) has more RAD51 foci than NTC control.
9. The figure 5f is not appropriate. the author should show the western blot results of USP48 and

RAD52 expression in the condition of figure 5e.

10. Where is the location of USP48 in cells? Some paper indicated the USP48 are major localized in cytosol (FASEB J. 2014 Mar; 28(3): 1422-34.) In figure 3A, the staining imaging indicated the USP48 looks like localized in nuclear.

11. The quality of Western blot of figure 3b, 4b and 5d are terrible, please improve. In addition, the molecular weight of siRNA-resistant wild-type (WT) or C98S mutant should be higher than endogenous USP48 due to containing FLAG Tag.

Response to Reviewers' comments.

Following this summary paragraph is a point-by-point response to each reviewer's comments. To summarise briefly our response we have undertaken all the experiments suggested by the reviewers and the results of these add considerable strength to our original submission. Broadly these main areas are:

We provide more evidence to support extended resection:

We have undertaken RPA foci analysis (Supplemental Fig S5F) and examined RAD51 intensity Supplemental Fig 5c, the results of these new experiments support the notion that loss of USP48 results in hyper-resection.

We provide more exploration of dependency of the observed extended resection in USP48 depleted cells on BRCA1 and on the resection machinery.

We include new data demonstrating:

- that the co-depletion of BRCA1 or CtIP with USP48 blocks the increase in recombination events measured by gene conversion and single strand annealing, seen on USP48 depletion alone (Supplemental Fig 9 a-c).
- that cell survival to Camptothecin seen following USP48 depletion requires BRCA1 and CtIP (Supplemental Fig 9d).
- that while Brca1 proficient mouse embryonic fibroblasts show an increase in the number of Rad51 foci formation following Usp48 depletion similar to that seen in HeLa and U2OS cells, MEFs that are Brca1 exon 11 mutant ($\Delta 11/\Delta 11$) (MEFs) do not show an increase in RAD51 foci following USP48 depletion (Supplemental Figure 6c-d).
- Similarly while Brca1 proficient mouse embryonic fibroblasts show an increase in camptothecin resistance following Usp48 depletion similar to that seen in HeLa cells, that Brca1 ^{$\Delta 11/\Delta 11$} MEFs fail to show an improvement in camptothecin resistance following Usp48 loss (Figure 9f).

These data strengthen the notion that in the absence of the ability to either initiate resection (loss of CtIP) or to lay down the H2Aub^{BRCA1} mark (loss of BRCA1), the silencing of USP48 cannot mediate hyper-resistance to CPT or hyper-recombination as judged by GFP-reporter assays for gene conversion and single-strand annealing.

We define spatial and temporal accumulation of USP48.

We have examined USP48 localisation with 53BP1 (Fig 3a) and established that USP48 accumulation at sites of DNA damage induced by microlaser requires BRCA1 itself (Fig 3a-b), placing the recruitment after that of BRCA1.

We have strengthened the phenotypes associated with H2A-Ub rescue of USP48-over-expression.

We have examined RAD51 foci formation to assess the effect of the H2Aub fusion following overexpression of EGFP-USP48 (Supplemental 7c&d). Consistent with the repressive impact of ectopic USP48 expression on resection (Fig 4d), overexpression of USP48 results in reduced RAD51 foci formation, moreover this can be rescued by expression of a H2A-ub fusion but not by WT H2A.

In addition, we have clarified the data regarding increased resection induced by the fusion making it clearer that indeed the fusion results in longer resection lengths (figure 4a). We have made a new H2A-Ub fusion construct with the K-R variants suggested and evaluated the new H2A-Ub for its ability to rescue the impact on USP48 over-expression (Supplemental Figure 7c & d). This new construct also rescues USP48 over-expression, supporting further our findings.

We include all the requested improved controls (SMARCD1 , CtIP and RAD52 western blots). In other comments where clarification of published, current data or phrasing was needed or a missed reference to be included we have made the changes needed. In addition we have explored potential for H2A site promiscuity by evaluation of over-expression on markers that reflect the presence of the K13/15 and polycomb marks in the DNA damage response, to address potential phenotypic similarity of USP48 with USP3.

During the reviewing process we also realized that we had initially focussed our efforts on isoform2, which lacks a C-terminal ubl domain. To ensure that our biochemical conclusions and rescue experiments were not caused by this difference, we have now repeated a substantial part of our experiments with isoform 1 (figures 1b-e, S1, 2a-f, S2 and S3a-c) and provide a comparison with isoform 2 in supplemental figure 4. Additionally, both isoforms are able to rescue the increased RAD51 foci DNA damage phenotype seen on USP48 depletion in cells (Supp Fig 5d-e).

The comments were tremendously useful in improving the clarity and strength of our findings and we thank the reviewers for their time and effort in the careful reading of our manuscript.

Reviewer #1 (Remarks to the Author):

BRCA1/BARD1 dimer have multiple roles in DNA double strand breaks repair. Some of them rely on its ubiquitin-ligase activity and in ubiquitination of specific targets. One of them is the ubiquitination of H2A, that is required to overcome the chromatin barrier during DNA resection. In this manuscript, the authors identify USP48 as a specific deubiquitinating enzyme of H2A, specifically the BRCA1-mediated ubiquitination, using an elegant in vitro approach. This deubiquitination works better when multiple lysines are conjugated, suggesting that the presence of one Ub stimulates the reaction.

In vivo, USP48 is recruited to DNA damage, and its depletion or lack of activity stimulates longer resection and RAD51 accumulation in BRCA1 and SMARCD1 dependent fashion. Mechanistically, affect the relocalization of 53BP1 within IRIF. As a consequence, lack of USP48 activity stimulates recombination and survival upon CPT treatment.

The experiments are well executed, and the conclusions are well supported by the experimental data. Overall, the story is well presented and interesting enough for Nat Commun audience. However, in my opinion, there are a few experiments that will strengthen the final model.

We thank the reviewer for their kind and thorough evaluation of the manuscript. In undertaking the experiments suggested we believe our evidence is stronger still to support the model.

1. No RPA foci are shown in the paper, but a measurement of resection length. Is this because USP48 depletion does not affect RPA foci formation (i.e. resection initiation)? This will fit the model, so RPA foci should be included as a supplementary figure.

In response to these comments we examined RPA foci and have now included new data showing the effect of USP48 depletion on RPA foci formation (Supplemental Fig S5F). These new data show an increase in RPA foci formation consistent with the increase seen in resection and RAD51 formation following USP48 depletion. Moreover the increase in RPA foci formation caused by USP48 depletion can be rescued by ectopic expression of WT but not the catalytically dead C98S form of USP48.

2. The in vitro study clearly shows the specificity of USP48 for H2A ubiquitination by BRCA1. The in vivo data support the idea as USP48 depletion is irrelevant for resection and Rad51 foci after BRCA1 downregulation. In that regard, it will strengthen the model if the hyper-recombination phenotype and hyper-resistance to CPT are also tested upon BRCA1 depletion. Along the same lines, one or both these phenotypes should be tested in the absence of CtIP, to demonstrate that rely on hyper-resection.

We agree, and have done the experiments suggested. The resulting data is as the reviewer predicted. Thus the hyper-recombination phenotype seen on USP48 depletion is lost on co-depletion of USP48 with either BRCA1 or CtIP (Supplemental Figure 9a-c). In addition, resistance to CPT seen upon USP48 depletion does not occur when USP48 depletion is combined with BRCA1 or CtIP depletions (Supplemental Figure 9d-e). Therefore, in the absence of the ability to either initiate resection (loss of CtIP) or to lay down the H2Aub^{BRCA1} mark (loss of BRCA1), loss of USP48 is unable to mediate hyper-resistance to CPT or hyper-recombination as judged by GFP-reporter assays for gene conversion and single-strand annealing. These new data are consistent with the suggested model of USP48 function.

3. In that regard, it is difficult to conclude from the data the contribution of H2A deubiquitination in the in vivo phenotypes. It is possible that USP48 deubiquitinates other BRCA1 substrates that affect resection, recombination and cell survival. I suggest to test some additional phenotypes in cells expressing an H2A-Ub and overexpressing USP48

This is an interesting point and we thank the reviewer for raising this concern. Certainly we agree that we cannot exclude the possibility that USP48 also deubiquitinates other substrates and possibly other BRCA1 substrates. The conclusion that H2Aub^{BRCA1} is the major USP48 substrate relevant to 53BP1 positioning and HR regulation is based on the evidence that expression of a H2A-ub fusion protein is sufficient to rescue the effects of USP48 overexpression on resection track lengths (Fig 4d and Supplemental 7c & d) and that the BRCA1-SMARCAD1 chromatin remodelling pathway is relevant to USP48 as shown by co-depletion and measurement of resection lengths (Fig 4A).

We have done as the reviewers suggested and now include new data where we have also examined RAD51 foci formation to assess the effect of the H2Aub fusion following overexpression of EGFP-USP48 (Supplemental 7c&d). Consistent with the repressive impact of ectopic USP48

expression on resection (Fig 4d), overexpression of USP48 results in reduced RAD51 foci formation, moreover this can be rescued by expression of a H2A-ub fusion but not by WT H2A. Thus while USP48 may de-ubiquitinate other BRCA1 substrates it appears that the supply of one, H2A-Ub, in a non-cleavable format can counter the repressive impact of USP48 over-expression in the relevant pathway of resection and RAD51 foci formation. Taken together these data suggest that if other USP48 substrates exist they either offer relatively minor roles to the biology we have examined, or at least the C-terminal H2A-Ub fusion is capable of overcoming the need for them.

4. Finally, in the model, the authors propose that USP48 limits resection by counteracting BRCA1 ubiquitination activity. Is this limitation a temporal or spatial? If is a temporal limitation, USP48 should be recruited later than BRCA1 to laser lines, something the authors can easily test using fluorescently tagged BRCA1 and USP48. If is a spatial restriction, the prediction is that it will show a similar accumulation pattern as 53BP1, measured in figure 4e. This will complete the model.

We agreed with the suppositions that the reviewer raised and have answered both questions. Firstly we find, as suggested, that USP48 closely co-localises with 53BP1 (Fig 3a). Secondly we examined the possible dependency of USP48's recruitment to sites of DNA damage and found that it requires BRCA1 itself (Fig 3a-b), placing the recruitment after that of BRCA1. We thank the reviewer for these suggestions!

Minor points:

The TAMRA gels look a bit fuzzy always in my copy of the manuscript. I am not sure is due to the conversion of the files, but it should be improved.

Supplementary figure S2a, the gels look fuzzy.

The fuzziness the reviewer is referring to might arise due to the detector settings of the fluorescence gel scanner chosen for these experiments. The pixel size under these settings is 50 um which makes the bands look fuzzy around the edges. Unfortunately we cannot provide a better resolution. We would like to point out that a higher resolution would not affect the quantification and the general conclusions drawn from these experiments, we apologize for the poor aesthetics

Reviewer #2 (Remarks to the Author):

This is an elegant study from the Sixma and Morris labs identifying the deubiquitinase USP48 as a regulator of H2A ubiquitination and thereby DNA end resection and DSB repair. The first part of the paper is biochemical – here, the authors identify USP48 as an H2A DUB with site-specificity for the BRCA1/BARD1 target sites on H2A and show an interesting mechanism of USP48 control by “auxiliary” ubiquitin-modifications. The second part of the paper is cell biological – here, the authors mainly characterize the phenotypes of an USP48 knock down. These phenotypes indicate

convincingly that a lack of USP48 leads to hyper-resection and consistently to hyper-recombination. Overall, this is a very high quality study to a biological phenomenon (DSB repair pathway choice), which is extremely important but still incompletely understood. As such this paper should be very interesting to the broad readership of Nature Communications. I only have very few points that should be addressed prior to publication.

We thank the reviewer for their kind evaluation of our work.

Major points:

1. Both parts of the paper are themselves convincing, but it would indeed be very nice to see a connection. I.e. does USP48 knock-down lead to enhanced levels of H2AK125,127,129ubi in damaged cells?

We agree that it would be very nice to have this data. We have tried to complete this experiment many times. As no reagents to the C-terminal H2A tail exist we have tried to approach this question using mutants of H2A. The experiments utilised a combination of H2A variants lacking K118/119, to preclude polycomb modification, with and without C-terminal lysines to indicate whether we could observe C-terminal modification on a background of USP48 loss or control siRNA treatment. These were co-transfected with his-tagged ubiquitin and performed as a nickel column purification to show covalent modification. Unfortunately the results of these assays have been uneven and we do not feel they are of sufficient quality for publication.

We hope the reviewer will agree that the body of evidence nevertheless now supports in cells the C-terminal specificity indicated by our *in vitro* experiments. These are:

- **USP48 over-expression experiments indicating N-terminal H2A-ub processing (sufficient to disrupt since 53BP1 accumulation, which is dependent on H2A-K13/15-ub, is not disrupted by USP48 overexpression (Supplemental Fig 7a)),**
- **evidence using BRCA1 and SMARCAD1 co-depletions and the impact on 53BP1 spread clearly shows that the impact of USP48 loss on resection requires the BRCA1-H2A-Ub driven pathway (Fig 4a-c, e)**
- **we have strengthened the phenotypic observations gleaned from our use of the H2-Ub fusion to build on the observations of rescue of resection lengths in USP48-over-expressing cells (fig 4d). We now also show that the loss of RAD51 foci induced by USP8 over expression can be rescued by this C-terminal H2A-Ub construct (Supplemental Fig 7c-d).**

2. Fig. 1b: The authors show values normalized to ubiquitin-rhodamine, probably because the relative activity of the different DUB preps are different. Nonetheless, it would be desirable to also show the non-normalized data.

The non-normalized data is shown in Supplemental Figure 1c and d. This is now more explicitly explained in the text and in figure legend 1b.

3. Fig. 3a: The USP48 localization to laser-induced foci is intriguing, but the authors should look at this localization in a time course experiment to establish whether USP48 localizes to DSB sites early

or late. Moreover, the finding of apparently endogenous USP48 foci is also intriguing and it would be good to know under which conditions (e.g. cell cycle state) USP48 forms foci in unchallenged cells.

To try to answer both of these queries, we have examined the whether the ability of USP48 to accrue to sites of damage is dependent on BRCA1. We find that it is (Fig 3a-b). The new data suggests, a time-frame of USP48 recruitment similar to, but after BRCA1, and a cell cycle stage in which BRCA1 recruitment is prevalent.

4. Fig. 4d: The experiment with the H2A-ubi fusion seems complicated to interpret, perhaps because the authors are using not only a covalent fusion between H2A and ubi, the constructs also lacks the lysines, which are normally targeted (as far as I can extract from the methods part). I am also puzzled why this fusion on the one hand counteracts the effect of USP48 overexpression, but does not enhance DNA end resection by itself. The authors should investigate this in more detail with a fusion containing the endogenous sites on H2A and another control mutant lacking the H2A lysines (is the effect of this mutant not best explained, if it counteracted both anti- and pro-resection factors?) or remove this part from the paper.

We can see that our experiment was confusing. As the reviewer expects the H2A-Ub (relabelled as H2A^{KR2}-Ub) alone DOES enhance resection. The median values for BrdU resection assay in Fig 4d are as follows:

NTC	35.2
Flag-USP48	17.8
H2A	35.7
H2A + FlagUSP48	17.8
H2A ^{KR2} -Ub	40.5
H2A ^{KR2} -Ub + FlagUSP48	37.6

This difference is supported statistically, as the Mann-Whitney test between NTC and H2A^{KR2}-Ub overexpression data sets is highly significant at $p < 0.005$. We apologise that this was not clearer in the original submission and have now included median values in the figures for all resection data, so that this small but highly significant difference is more obvious.

We have also included new data using the same H2A^{KR2}-Ub fusion in Supplemental Fig 7C & D (in response to concerns raised by reviewer 1). These data show that expression of the H2A^{KR2}-Ub fusion alone increases RAD51 foci formation, consistent with the increase we see in resection, and with the expectations of the reviewer.

Nevertheless to avoid possible artefact from the mutant H2A (which indeed lacks many lysines) we have also examined the impact of a H2A-ub fusion that is only missing the K125/127/129R site in order to assess the contribution of the Ub-fusion in the absence of the BRCA1 site, but retaining lysines elsewhere as the reviewer suggested (labelled H2A^{KR1}-Ub). We have assessed this fusion as well as the fusion lacking all lysines associated with the DNA damage response for the ability to rescue RAD51 foci formation in cells over-expressing USP48 (Supplemental Fig 7 c & d). We find that both have the ability to restore high RAD51 foci numbers, suggesting that the fused ubiquitin, rather than the mutant lysines, is the dominant effector of the rescue.

5. Fig. 3-4: The authors use number of Rad51 foci as measure of DNA end resection and HR. Have they thought of using foci intensity as additional measure of resection.

This was a lovely suggestion. We have now included new data in Supplemental Fig 5c which, as the reviewer anticipated, shows that RAD51 foci intensity is also increased following USP48 depletion.

Minor points:

1. Fig. 5c: This experiment indicates that hyperresection may lead to camptothecin resistance, which is intriguing. The Pfander lab has recently reported that in yeast hyperresection leads to camptothecin resistance (Bantele et al. 2017, eLIFE). I would encourage the authors to discuss this study, as both conditions apparently hyperactivate resection via the SMARCAD1/Fun30 pathway.

We apologise to the reviewer for missing this paper as we were aware of this interesting story from the Pfander lab and fully intended to cite it in the original submission. This is an oversight from us and we have now included discussion on this study.

2. Even though the writing is very clear throughout the paper, there are a few sentences/expressions that are somewhat unclear and should be changed.

abstract & end of p2, l69: “auxiliary ubiquitin” – this is only defined later in the paper and remains unclear at first

We apologize for the lack of clarity. We have now added a brief description of the auxiliary ubiquitin to the abstract and introduction.

abstract: “extended 53bp1 positioning”

We have rephrased this sentence

p2, l52: “synthetic viability”

We have rephrased this sentence

p5, l165: “track-length of ... were spread an measured”

We have rephrased this sentence

p5, l178: “Resection promoted by the BRCA1-BARD1 E3 Ub ligase promotes SMARCAD1-mediated remodelling of chromatin-associated 53BP1.”

We have rephrased this sentence

3: Fig. 2a: The stacking of the westerns looks strange to me. If the authors want to show this data they should find a different layout.

The stacked fluorescence scans of the gels in Figure 2a represent four selected conditions from all the different conditions used for the kinetic modelling (all conditions used are shown in Figure S2a). By using this type of layout for Figure 2a we hope to illustrate the dynamic nature and the range of conditions used for fitting the kinetic model. We have altered the text to make it more explicit that these conditions are chosen as an example and now provide clearer cross-references to the whole data set in Figure S2a.

4. Fig. 4b: The SMARCAD1 western for the siRNA control does not look very convincing. It would be good to have a better (and perhaps quantified) western. If not possible please provide an mRNA-based expression control.

We now include a new western blot for the SMARCAD1 siRNA control western blot. This is shown in Fig 4B.

Reviewer #3 (Remarks to the Author):

While the impact of BRCA1 in orchestrating DNA damage repair and tumor susceptibility is highly recognized, the role of BRCA1/BARD1 acting as ubiquitin protein ligase has been controversial (Ludwig T. et al. Science vol 334, 2011 and Jonkers J. et al. Cancer Cell 20, 2011). Previously report described the ubiquitin ligase activity for BRCA1/BRAD1 (Wu L. et al. Nat. Genet. 14, 1996 and Hashizume R. et al. JBC 276, 2001). It was thought that H2A is one of the major targets of BRCA1/BARD1 in response to DNA damage through K125/127/129 on H2A. Moreover, the author revealed that SMARCAD1 is a cascade down-stream of H2A ubiquitination to 53BP1 positioning and resection. In this manuscript, the authors deciphered the role USP48 to be a deubiquitinating enzyme that counteracts the BRCA1/BARD1-catalized H2A ubiquitination and further linked the function of USP48 to regulating DNA end *resection* and RAD51 recruitment. While addition of USP48 to the paradigm of BRCA1/BARD1-H2A-DNA end resection/RAD51 recruitment is new, the data present at this time is not enough to prove the hypothesis.

We thank the reviewer for the appreciation of the novelty and hope that the new data we provide in response to their comments and suggestions now present enough information to support the proposed hypothesis.

Following are concerns:

1. The USP3 has the same activity as USP48 on deubiquitinating of H2ABRCA1Ub. Therefore, whether USP3 has a compensative effect on H2ABRCA1Ub modification, gene conversion and single strand annealing when USP48 is knockdown? Which one is the major regulator of H2ABRCA1Ub deubiquitination.

We think this is a misunderstanding by the reviewer. USP3 and USP48 only seem to have the same activity when normalized to their respective activity on minimal substrate ubiquitin rhodamine. To compare the activity of both enzymes on a nucleosomal substrate directly the data shown in figure S1c needs to be considered. This shows that USP48 has a roughly 10-fold higher activity on H2A^{BRCA1ub} than USP3. Taken into consideration that when using USP48 isoform 1 we have to use 10-fold less enzyme in these assays the real difference in activity could easily be 100-fold.

Furthermore, to answer the query about redundancy, we observe the phenotypes of USP48 knockdown despite USP3 being present, which would argue against a compensatory effect. It is clear that depletion of USP48 alone has a strong and distinctive impact on biology and, therefore, this does not suggest redundancy with another DUB (i.e. another DUB does not compensate for loss of USP48 and mask this effect).

To directly address the possibility that USP48 may have a similar role to USP3 we performed over-expression studies similar to those previously performed for USP3. USP3 can cleave Ub from H2A and previous publications have shown its over-expression¹ (or that of USP44 which has the same activity²) reduce 53BP1 and BRCA1 accumulation. This is most likely due to their processing of H2Aub modification (K118/119 and K13/15).

We show that overexpression of EGFP-USP48^{iso1}, unlike the published phenotype of USP3 and USP44 overexpression, does not affect either 53BP1 or BRCA1 foci formation (Supplemental Fig S7a & b). Since 53BP1 foci formation relies on H2A-K13/15ub, and recruitment of both 53BP1 and BRCA1 to damage foci is lost following inhibition of RING1B/BMI1^(3,4) this new data suggests USP48 is unlikely to behave as a DUB able to generally remove Ub mark from all Ks of H2A in cells.

2. The approach bridging USP48 to H2A ubiquitination sounds candidate-driven manner.

The reviewer is right in stating this. We initiated this study with the aim of finding deubiquitinating enzymes that show specific cleavage of H2A and we pre-selected DUBs that we deemed likely to cleave H2A. The approach is indeed candidate driven. When we tested DUB candidates that interact with H2A, we found that one of these acts on the BRCA1 H2A-ub site and pursued our study accordingly.

3. The authors need to present whether USP48 promotes genome stability depend on BRCA1-H2A or not in BRCA1 mutant or loss cells.

The comments prompted us to perform several experiments to test the dependency on BRCA1 further. Thus in addition to our demonstration that when BRCA1 is depleted from cells the further loss of USP48 has little impact on resection (Figure 4a) or RAD51 foci formation (Figure 4c), we now include new data demonstrating:

1. that the co-depletion of BRCA1 and USP48 blocks the increase in recombination events measured by gene conversion and single strand annealing assays, seen on USP48 depletion alone (Supplemental Fig 9 a-c).

2. that cell survival to Camptothecin seen following USP48 depletion requires BRCA1 (Supplemental Fig 9d).
3. that while Brca1 proficient mouse embryonic fibroblasts shows an increase in the number of Rad51 foci formation following Usp48 depletion similar to that seen in HeLa and U2OS cells, MEFs that are Brca1 exon 11 mutant ($\Delta 11/\Delta 11$) do not show an increase in Rad51 foci following Usp48 depletion (Supplemental Figure 6c-d).
4. that while Brca1 proficient mouse embryonic fibroblasts show an increase in camptothecin resistance following Usp48 depletion, similar to that seen in HeLa cells, that Brca1 ^{$\Delta 11/\Delta 11$} fail to show an improvement in camptothecin resistance following Usp48 loss (Figure 9f).

4. The author showed USP48 loss resulting in increased end resection and RAD51 foci numbers. Whether modulating of USP48 expression affects the numbers, size, and intensity of single-strand DNA binding protein RPA loading on the resected ssDNA?

We thank the reviewer for their comments and have now included new data showing the effect of USP48 depletion on RPA foci formation (Supplemental Fig 5f). These new data show an increase in RPA foci formation consistent with the increase in resection and RAD51 formation seen following USP48 depletion. The increase in RPA foci formation caused by USP48 depletion can be rescued by ectopic expression of WT but not the catalytically dead C98S form of USP48.

5. The colony survival assay has only been investigated in HeLa cells. Whether modulating of USP48 expression affect colony survival in BRCA1 mutant or loss cells? Whether the effect of USP48 expression on cell survival is depend on counter-act on BRCA1-H2A ubiquitination cascade?

In response to this query we have tested the effects of USP48 co-depletions with BRCA1 on the cell survival hyper-resistance to CPT phenotype. These new data are shown in Supplemental Fig 9d. The resistance to CPT seen upon USP48 depletion does not occur when USP48 depletion is combined with BRCA1 depletion. In response to the concern about cell type we have also confirmed our findings in mouse embryonic fibroblasts, showing that while Brca1 proficient mouse embryonic fibroblasts show an increase in camptothecin resistance following Usp48 depletion similar to that seen in HeLa cells, that Brca1 ^{$\Delta 11/\Delta 11$} fail to show an improvement in camptothecin resistance following Usp48 loss (Figure 9f). Therefore, in both human and mouse settings it appears that in the absence of the ability to lay down the H2Aub^{BRCA1} mark (loss of BRCA1), that subsequent loss of USP48 is unable to mediate hyper-resistance to camptothecin, supporting the model.

6. This manuscript chooses the list of DUBs in the DNA damage response (USP1/UAF1, USP11, USP7, USP15, USP12/UAF1) from Nat Cell Biol. 2014 Oct; 16(10): 1016–8. However, USP48 is not in the list of DUBs which have the features that localization of GFP–DUBs to laser micro-irradiation sites and effects on DSB repair using siRNA pools (neutral comet assay), but in the list of DUBs which G2/M affect G2/M checkpoint.

We believe this is a misunderstanding as the reviewer is only partially correct. USP48 was not tested in the study referenced by the reviewer and hence it is not in their list of DUBs that recruit to sites of damage. As stated in our manuscript, we chose USP48 because another study showed binding of USP48 to H2A (Kalb, R. *et al. Nat. Struct. Mol. Biol.* 21, 569–71 (2014)⁵).

7. In figure S4d, knockdown 53BP1 seems to promote Rad51 foci formation. This is a conflict with previous studies (Nucleic Acids Res. 2013 Nov;41(21):9719-31; Nat Struct Mol Biol. 2010 Jun;17(6):688-95.).

In the NSMB – Bowman paper referred to Rad51 foci are quantified on a percent of cells with >10 foci and the number of foci per cell were not addressed. This method of quantification is common across the field and masks the actual numbers of Rad51 foci per cell.

The NAR-2013⁶ paper indeed sees no increase in RAD51 after IR on 53BP1 depletion. It should be noted that the NAR paper is the exception. In contrast the majority of published work has suggested loss of 53BP1 increases homologous recombination (to the extent that some call it an ‘anti-recombinase’ (eg Callen *et al*⁷)). Examples in which 53BP1- or 53BP1 effector proteins increase RAD51 or measures of HR include: (PTIP – RAD51⁷) Feng⁸, (RIF1 increase in % cells with RAD51 and increase in HR). Moreover we see the increase in RAD51 foci per cell (stained for s-phase cell cycle markers), reproducibly. For other examples see Densham *et al* NSMB 2016⁹, further we see the same impact in mouse embryonic fibroblasts in which 53bp1 is genetically ablated (data herein for the reviewer). Thus this observation is not specific to the system we have used in the manuscript but common across many cell types and from many research groups.

8. The overexpressed FLAG-USP48 protein is about several folds of the endogenous USP48 (fig 3b, 5c), but in fig 3e, FLAG-USP48 (WT) has more RAD51 foci than NTC control.

We thank the reviewer for their comments and have revisited this experiment. We now include an additional experimental data set into the RAD51 count following USP48 depletion and re-

expression of either Flag-WT or C98S USP48 (updated supplemental fig 5e). In addition we have also re-visited the corresponding western blots for this experiment (as asked for by Reviewer 3 in point 11). These are now in agreement (Fig 3c).

9. The figure 5f is not appropriate. the author should show the western blot results of USP48 and RAD52 expression in the condition of figure 5e.

We have now obtained a RAD52 antibody that is able to detect endogenous RAD52 from cell lysates (gift from Fena Ochs/Claudia Lukas, Copenhagen as not commercially available). Figure 5e now shows the western blot corresponding to USP48, RAD51 and RAD52 co-depletions.

10. Where is the location of USP48 in cells? Some paper indicated the USP48 are major localized in cytosol (FASEB J. 2014 Mar;28(3):1422-34.) In figure 3A, the staining imaging indicated the USP48 looks like localized in nuclear.

We have examined endogenous USP48 localisation using an antibody to endogenous USP48. We detect nuclear and nuclear foci localisation which is lost on USP48 depletion (see new Fig 3A – laser lines for endogenous USP48 following control or USP48 siRNA in 4% PFA fixed cells). There is clear recruitment of USP48 to sites of laser-induced damage (Fig 3A) irrespective of the localisation of endogenous USP48 in undamaged cells.

Indeed we also find that exogenous EGFP-USP48^{iso1} is nuclear, while exogenous Flag-USP48^{iso2} is largely cytoplasmic but not exclusively so, with both nuclear and cytoplasmic localisation seen in a small percentage of cells. In either case, expression of either USP48^{iso1} or USP48^{iso2} is able to rescue the DNA damage phenotypes of USP48 depletion. Indeed, while the differing cellular localisations of USP48 isoforms is intriguing and raises possibilities of differential isoform expression in different cell types – this is something we have not explored in detail for the purposes of this manuscript as both isoforms are proficient at rescuing the DNA damage phenotypes seen on USP48 depletion.

11. The quality of Western blot of figure 3b, 4b and 5d are terrible, please improve. In addition, the molecular weight of siRNA-resistant wild-type (WT) or C98S mutant should be higher than endogenous USP48 due to containing FLAG Tag.

We apologise for the western blot quality and now include new improved westerns for Fig 3B (now the new Fig 3c), Fig 4B and Fig 5D as requested.

While we appreciate that adding a FLAG-tag increases the size of the tagged protein, we would not expect to be able to resolve that difference in our blots as the difference is too small in kDa to detect (only an additional 1 kDa (FLAG) on a protein that normally runs at just under 130 kDa).

References.

1. Sharma, N. et al. USP3 counteracts RNF168 via deubiquitinating H2A and gamma H2AX at lysine 13 and 15. *Cell Cycle* 13, 106-114 (2014).

2. Mosbech, A., Lukas, C., Bekker-Jensen, S. & Mailand, N. The deubiquitylating enzyme USP44 counteracts the DNA double-strand break response mediated by the RNF8 and RNF168 ubiquitin ligases. *The Journal of Biological Chemistry* 288, 16579-87 (2013).
3. Ismail, I.H., Andrin, C., McDonald, D. & Hendzel, M.J. BMI1-mediated histone ubiquitylation promotes DNA double-strand break repair. *The Journal of cell biology* 191, 45-60 (2010).
4. Ismail, I.H., McDonald, D., Strickfaden, H., Xu, Z. & Hendzel, M.J. A small molecule inhibitor of polycomb repressive complex 1 inhibits ubiquitin signaling at DNA double-strand breaks. *The Journal of Biological Chemistry* 288, 26944-54 (2013).
5. Kalb, R., Mallery, D.L., Larkin, C., Huang, J.T. & Hiom, K. BRCA1 is a histone-H2A-specific ubiquitin ligase. *Cell reports* 8, 999-1005 (2014).
6. Kakarougkas, A. et al. Co-operation of BRCA1 and POH1 relieves the barriers posed by 53BP1 and RAP80 to resection. *Nucleic acids research* 41, 10298-311 (2013).
7. Callen, E. et al. 53BP1 mediates productive and mutagenic DNA repair through distinct phosphoprotein interactions. *Cell* 153, 1266-80 (2013).
8. Feng, L., Fong, K.W., Wang, J., Wang, W. & Chen, J. RIF1 counteracts BRCA1-mediated end resection during DNA repair. *J Biol Chem* 288, 11135-43 (2013).
9. Densham, R.M. et al. Human BRCA1-BARD1 ubiquitin ligase activity counteracts chromatin barriers to DNA resection. *Nat Struct Mol Biol* 23, 647-655 (2016).

Reviewers' Comments:

Reviewer #1:

Remarks to the Author:

I have revised this new version of the manuscript with great interest and pleasure. The authors have addressed all my suggestions fully. In my opinion they have improved an already good story. Hence, I fully support its publication in Nat Communications as it is.

Reviewer #2:

Remarks to the Author:

"USP48 restrains resection by site specific cleavage of the BRCA1 ubiquitin mark from H2A." by Michael Uckelmann, Ruth M. Densham, Herrie H. K. Winterwerp, Alexander Fish, Titia K. Sixma, and Joanna R. Morris

The authors did address my concerns with the revised version of their manuscript, which I find further improved.

Additionally, however, the authors realized during the revision of this paper that the USP48 version (isoform 2) initially used in their study lacked an UBL domain. Given the known, important roles of UBL domains in the function of deubiquitinases, this is an obvious concern, but was only to a certain extent addressed by the authors. Nonetheless, I would strongly urge the authors to further clarify this critical point before publication. I see two possible ways to go forward.

(1) show that the initially (and throughout the paper) used isoform 2 (rather than isoform 1) is the physiological relevant isoform and concentrate on this isoform in the paper.

OR

(2) complete their story with the missing experiments for isoform 1. This holds particularly true for the in vivo experiments in the second part of the paper. I think that all experiments that are critical for the main conclusion of the paper need to be repeated and show both isoforms side-by-side (Rad51 foci analysis, BrdU-based resection assay and the CPT sensitivity assays, potentially also the recombination assays).

Under these exceptional circumstances, I would advise to give the authors more time to further revise their paper.

Reviewer #3:

Remarks to the Author:

The authors addressed all questions raised adequately. Results from the suggested experiments now support the hypothesis and conclusion in good shape. Thus, the quality of this manuscript is now qualified.

Response to Reviewers' comments:

Reviewer #1 (Remarks to the Author):

I have revised this new version of the manuscript with great interest and pleasure. The authors have addressed all my suggestions fully. In my opinion they have improved an already good story. Hence, I fully support its publication in *Nat Communications* as it is.

We thank the reviewer for their comments.

Reviewer #2 (Remarks to the Author):

“USP48 restrains resection by site specific cleavage of the BRCA1 ubiquitin mark from H2A.”
by Michael Uckelmann, Ruth M. Densham, Herrie H. K. Winterwerp, Alexander Fish, Titia K. Sixma, and Joanna R. Morris

The authors did address my concerns with the revised version of their manuscript, which I find further improved.

Additionally, however, the authors realized during the revision of this paper that the USP48 version (isoform 2) initially used in their study lacked an UBL domain. Given the known, important roles of UBL domains in the function of deubiquitinases, this is an obvious concern, but was only to a certain extent addressed by the authors. Nonetheless, I would strongly urge the authors to further clarify this critical point before publication. I see two possible ways to go forward.

(1) show that the initially (and throughout the paper) used isoform 2 (rather than isoform 1) is the physiological relevant isoform and concentrate on this isoform in the paper.

OR

(2) complete their story with the missing experiments for isoform 1. This holds particularly true for the *in vivo* experiments in the second part of the paper. I think that all experiments that are critical for the main conclusion of the paper need to be repeated and show both isoforms side-by-side (Rad51 foci analysis, BrdU-based resection assay and the CPT sensitivity assays, potentially also the recombination assays).

Under these exceptional circumstances, I would advise to give the authors more time to further revise their paper.

We were also concerned about the discovery that our initial USP48 isoform lacked part of the C-terminal UBL domain. In the previous version of the paper we had evaluated the impact of both USP48^{iso1} and USP48^{iso2} expression on the ability to impact RAD51 foci numbers in cells treated with USP48 siRNA. We saw that both USP48^{iso1} and USP48^{iso2} restore RAD51 foci numbers to the same

degree, and the RAD51 foci numbers seen are similar to those levels observed in control cells (Supplemental Fig 5 d and e).

Now we have further addressed the USP48^{iso1} and USP48^{iso2} forms in multiple assays to address the concern expressed that the UBL domain might bring activity in cells in the following new ways:

1. We have evaluated the impact of both USP48^{iso1} and USP48^{iso2} expression on the ability to shorten resection lengths in cells treated with USP48 siRNA (siRNA sequences used in the study are able to target all catalytically active forms of USP48). These data show both USP48^{iso1} and USP48^{iso2} restore section lengths to the same degree, back to the lengths observed in control cells (Fig 3 c-f).
2. We have evaluated the impact of both USP48^{iso1} and USP48^{iso2} expression on the ability to impact Camptothecin resistance in cells treated with USP48 siRNA. Both USP48^{iso1} and USP48^{iso2} restore camptothecin sensitivity to the same degree, similar to the levels observed in control cells (Fig 5 c and d).
3. We have evaluated the impact of both USP48^{iso1} and USP48^{iso2} expression on the ability to impact RPA foci numbers in cells treated with USP48 siRNA. Both USP48^{iso1} and USP48^{iso2} restore RPA foci numbers to the same degree, to the numbers observed in control cells (Supplemental Fig 5 f and g).

Thus while isoform 1 is more active *in vitro*, the lack of the UBL in isoform 2 is clearly not detrimental to the regulation of resection in cells. I believe this answers the reviewers concern about the possible role of the UBL domain, addressing measures of resection itself and the impact on Camptothecin sensitivity.

Moreover we have improved the manuscript to make the findings (Iso 1 and Iso 2) more prevalent as their equivalence in the cell experiments may have been missed.

Reviewer #3 (Remarks to the Author):

The authors addressed all questions raised adequately. Results from the suggested experiments now support the hypothesis and conclusion in good shape. Thus, the quality of this manuscript is now qualified.

We thank the reviewer for their comments.

Reviewers' Comments:

Reviewer #2:

Remarks to the Author:

"USP48 restrains resection by site specific cleavage of the BRCA1 ubiquitin mark from H2A." by Michael Uckelmann, Ruth M. Densham, Herrie H. K. Winterwerp, Alexander Fish, Titia K. Sixma, and Joanna R. Morris

The authors did address my concerns regarding a potential difference between USP48 isoforms 1 and 2 and a potential impact and have also put sufficient emphasis on this issue in the revised version of the manuscript. The new data convincingly shows that the USP48 appears not to play a critical role in the regulation of the DSB response in vivo.

As such the authors should be congratulated on an excellent piece of work that in my view is ready for publication in Nature Comm.